# Instability in Diffusion ODEs: An Explanation for Inaccurate Image Reconstruction

## Abstract

Diffusion reconstruction plays a critical role in various applications such as image editing, restoration, and style transfer. In theory, the reconstruction should be simple, as it just inverts and regenerates images by numerically solving the Probability-Flow Ordinary Differential Equation (PF-ODE). Yet in practice, noticeable reconstruction errors have been observed, which cannot be well explained by numerical discretization error alone. In this work, we identify an intrinsic property of the PF-ODE generation process, the *instability*, that can further amplify the reconstruction errors. The root of this instability lies in *the sparsity inherent in the generation distribution: the probability mass is concentrated on scattered small regions, while most of the space remains nearly empty.* To demonstrate the existence of instability and its amplification on reconstruction error, we conduct experiments on both toy numerical examples and popular open-source diffusion models. Furthermore, based on the characteristics of image data, we theoretically prove that the probability of instability converges to one as the data dimensionality increases. Our findings clarify that instability, besides numerical errors, is a fundamental cause of inaccurate diffusion reconstruction, and offer insights for future improvements.

## 1 Introduction

Diffusion models have rapidly emerged as a pivotal class of generative models, demonstrating superior performance, particularly in image generation (Ho et al., 2020; Rombach et al., 2022; Saharia et al., 2022; Ramesh et al., 2022; Balaji et al., 2023; Pernias et al., 2023; Peebles & Xie, 2023; Podell et al., 2023; Kawar et al., 2023; Li et al., 2023; Esser et al., 2024; Liu et al., 2024). A fundamental technique within diffusion models is diffusion reconstruction, which comprises diffusion inversion (Song et al., 2020; Ramesh et al., 2022; Chung et al., 2022b) and diffusion generation–both executed through numerical solving the Probability Flow Ordinary Differential Equations (PF-ODEs) (Song et al., 2021; Lu et al., 2022a). Diffusion inversion[1] first converts an image into an inverted noise, which is then used by diffusion generation process to reconstruct the original image. Diffusion reconstruction is crucial due to its extensive applications in downstream tasks, including image editing (Gal et al., 2022; Hertz et al., 2022), restoration (Xiao et al., 2024), and style transfer (Su et al., 2022). In editing scenarios, the goal is not only to inject new information but also to preserve existing features. This makes exact reconstruction a fundamental prerequisite for reliable downstream use.

Diffusion reconstruction appears straightforward: one simply integrates the PF-ODE forward to obtain the inverted noise and then integrates backward to regenerate the original image. Yet in practice, significant reconstruction errors can occasionally happen, as shown in Figure 1. While previous works largely attribute these inaccuracies to numerical discretization of ODE solvers (Wallace et al., 2023; Wang et al., 2024; Lin et al., 2024; Zhang et al., 2024), such explanations are insufficient to account for the substantial discrepancies that are sometimes visually perceptible. This gap calls for a different explanation.

In this work, we point out the existence of *instability* in the diffusion generation process, and demonstrate its amplification effect on the reconstruction errors. Specifically, instability characterizes sce-

---

[1]We follow the convention that uses *inversion* indicating solving PF-ODE along the forward time (Mokady et al., 2023; Wallace et al., 2023; Zhang et al., 2024).

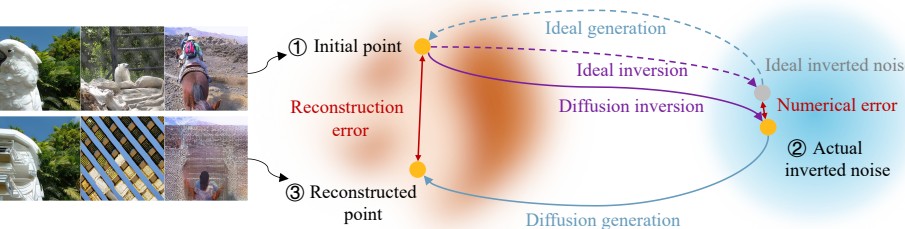

Figure 1: **Instability amplifies diffusion reconstruction errors**. Given an *initial point* in the *data distribution*, the reconstruction process first undergoes *diffusion inversion* to produce the *actual inverted noise*. *Numerical errors* introduced by the inversion will cause the actual inverted noise to deviate from the theoretically ideal inverted noise. When the generation process exhibits instability, these numerical errors are amplified, resulting in significantly larger *reconstruction errors*. This amplification makes accurate reconstruction challenging.

narios where the diffusion generation process is sensitive to the initial noise. Thus, any numerical error in the inverted noise will be amplified during the regeneration phase. Figure 1 illustrates the process by which reconstruction errors arise.

To further demonstrate the presence of instability from a theoretical perspective, we establish a lower bound on the probability of instability. Note that we analyze the ideal diffusion reconstruction process, free from numerical errors, to isolate properties of PF-ODE reconstruction that do not depend on the choice of numerical solver. Here, inherent is meant in this solver-agnostic sense—independence from discretization errors. Based on reasonable assumptions about the real image distribution and the generation distribution of diffusion models, we demonstrate that when the data dimensionality increases as infinity, the probability of instability in reconstruction tends to one! Considering the high dimensionality of image data, such surprising asymptotic result provides theoretical justification for the instability observed in image reconstruction within diffusion models.

**Mechanisms of instability.** For better understanding on the mechanism behind the instability in diffusion reconstruction, we provide an intuitive illustration in Figure 2. Our analysis reveals that *the sparsity of the generation distribution plays a pivotal role in the emergence of instability*. Here, the sparsity means that the generation distribution is concentrated in scattered, small regions, while the majority of regions possess low probability density. Recall that the PF-ODE-based generation process actually builds a push-forward mapping from the Gaussian distribution to generation distribution. According to the push-forward formula, the generation mapping must preserve probability. Thus, *suppose a region $A$ in the Gaussian distribution is mapped to a significantly lower density region $B$ in the generation distribution, the area of region $B$ will be extended to maintain the probability*. This expansion necessitates large gradients in the generation process, indicating the emergence of instability, since local area enlargement directly corresponds to large Jacobian norms and thus heightened sensitivity to perturbations. Meanwhile, during image reconstruction, the initial image is sampled from some underlying real distribution. This real distribution is generally different from the generation distribution. Such distribution discrepancy can lead to a non-negligible probability that the initial image resides in the low density region of the generation distribution, and thus make the instability high probable.

**Main organization.** In Section 2, we will introduce preliminaries about PF-ODE and the definition of instability. Subsequently in Section 3, we will provide experimental evidence that instability actually exists in diffusion generation, and then demonstrate its amplification on reconstruction error. Finally in Section 4, we further provide theoretical evidence on instability. Based on the characteristics of image distributions, we reveal that the sparsity of generation distribution can induce the instability, and theoretically prove that the instability will almost surely occur during reconstruction for infinite dimensional images.

## 2 PRELIMINARIES

### 2.1 DIFFUSION MODELS AND PROBABILITY FLOW ODE

Diffusion models can generate samples under desired distribution $\pi_{\text{gen}}$ from noise under standard Gaussian distribution $\mathcal{N}(\mathbf{0}, \mathbf{I})$ (Ho et al., 2020; Song et al., 2021). The generation process can be

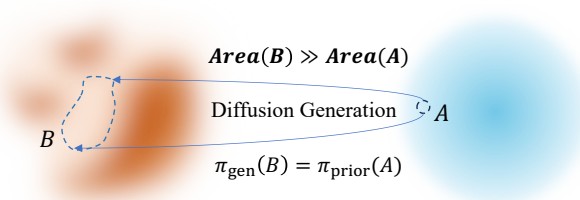

$$Area(B) \gg Area(A)$$

Diffusion Generation

$B$

$A$

$$\pi_{\text{gen}}(B) = \pi_{\text{prior}}(A)$$

Generation Distribution: $\pi_{\text{gen}}$      Gaussian: $\pi_{\text{prior}}$

Figure 2: **Intuitive illustration of instability occurrence during the generation process.** For typical image data, *the generation distribution is inherently sparse, meaning that most of the probability mass is concentrated in scattered, small regions, while the majority of the space has low probability.* In contrast, the prior Gaussian distribution concentrates its probability within a bounded region and is relatively uniformly distributed. According to the push-forward formula, *the generation mapping must preserve probability measures, ensuring that the image of a probability region $A$ under the generation mapping $B$ satisfies $\pi_{\text{gen}}(B) = \pi_{\text{prior}}(A)$.* When $B$ falls into a low-density region of the generation distribution, significantly lower than the density at $A$, maintaining probability equality requires that the area (or more rigorously, the Lebesgue measure) of $B$ be much larger than that of $A$. This amplification effect results in large gradients during the generation process, indicating the emergence of instability. The probability of the instability is analyzed in Section 4 in detail.

achieved by numerically solving the PF-ODE (Anderson, 1982; Song et al., 2021; Liu et al., 2023):

$$d\boldsymbol{x}_t/dt = \boldsymbol{v}(\boldsymbol{x}_t, t), \text{ where } \boldsymbol{v}(\boldsymbol{x}, t) = \mathbb{E}[Z - X|X_t = \boldsymbol{x}], \quad (1)$$

$t \in [0,1]$, $\boldsymbol{v} : \mathbb{R}^n \times [0,1] \to \mathbb{R}^n$ is a time-dependent vector field defined by the conditional expectation in Equation (1). In the expectation, the noise $Z \sim \mathcal{N}(\boldsymbol{0}, \mathbf{I})$ and the data $X \sim \pi_{\text{gen}}$ are independently sampled, $X_t = (1-t)X + tZ$ is a linear interpolation between them.

Note that here we adopt the flow matching formulation to represent the PF-ODE (Liu et al., 2023; Lipman et al., 2023), and the subsequent sections will consistently utilize this formulation. Here, we adopt the flow matching formulation for two reasons: 1) the flow matching formulation is actually equivalent to conventional PF-ODEs when one distribution is set to Gaussian (Song et al., 2021; Liu et al., 2023; Albergo et al., 2023), and 2) this formulation is widely adopted in recent top-tier text-to-image models (Esser et al., 2024; Black Forest Labs, 2024).

**Diffusion generation and inversion.** Both diffusion generation and inversion solve the PF-ODE in Equation (1), but in opposite directions. Diffusion generation mapping can produce data samples from initial noise, while diffusion inversion mapping yields an inverted noise from an initial data. We formally define these two mappings as follows:

**Definition 2.1** (Diffusion generation and inversion mappings)**.** The diffusion generation mapping $G : \mathbb{R}^n \to \text{supp}(\pi_{\text{gen}})$ is defined as $G(\boldsymbol{z}) = \boldsymbol{x}_0$, where $\boldsymbol{x}_0$ is the solution at $t = 0$ of the ODE Equation (1) with the initial value $\boldsymbol{x}_1 = \boldsymbol{z}$ at $t = 1$. Meanwhile, the diffusion inversion mapping can be denoted by $G^{-1}$, the inverse function of the generation mapping.

## 2.2 DEFINITION OF INSTABILITY

For later analysis, we first provide the definition of Intrinsic instability for general mapping $F$.

**Definition 2.2** (Intrinsic instability)**.** Suppose $F : \mathbb{R}^n \to \mathbb{R}^n$ is a continuously differentiable mapping. For an input vector $\boldsymbol{x} \in \mathbb{R}^n$, if there exists another non-zero vector $\boldsymbol{u} \in \mathbb{R}^n$ such that

$$\mathcal{E}_F(\boldsymbol{x}, \boldsymbol{u}) := \|J_F(\boldsymbol{x})\boldsymbol{u}\|/\|\boldsymbol{u}\| > 1, \quad (2)$$

where $J_F(\boldsymbol{x}) \in \mathbb{R}^{n \times n}$ denotes the Jacobian matrix of $F$ evaluated at $\boldsymbol{x}$, then we say that $F$ exhibits the *intrinsic instability* at $\boldsymbol{x}$ in the direction $\boldsymbol{u}$. The scalar $\mathcal{E}_F(\boldsymbol{x}, \boldsymbol{u})$ is referred to as the *intrinsic instability coefficient* of $F$ at $\boldsymbol{x}$ along the direction $\boldsymbol{u}$.

When the intrinsic instability coefficient exceeding one, it indeed indicates that the mapping $F$ amplifies any *infinitesimal perturbation* along the specific direction $\boldsymbol{u}$. The magnitude of $\mathcal{E}_F(\boldsymbol{x}, \boldsymbol{u})$ quantitatively measures the sensitivity of $F$ to changes in the input direction $\boldsymbol{u}$. The following

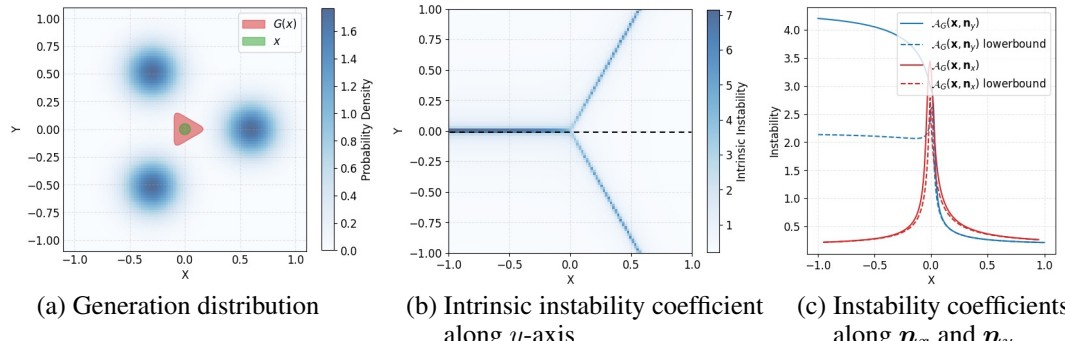

(a) Generation distribution     (b) Intrinsic instability coefficient along $y$-axis     (c) Instability coefficients along $\boldsymbol{n}_x$ and $\boldsymbol{n}_y$

Figure 3: **Experimental evidence on the existence of instability for a two-dimensional generation distribution.** (a) visualizes the density function of the generation distribution, which is a mixture of Gaussians generated from a standard gaussian. The green region, after applying generation mapping, becomes a significantly larger red region. (b) visualizes the intrinsic instability coefficient along the $y$-axis. It is observed that the intrinsic instability coefficient along the $y$-axis in the central region significantly exceeds one, consistent with the position of the green region in (a). For each point on the dashed line in (b), we compute the instability coefficient $\mathcal{A}_G(\boldsymbol{x}, \boldsymbol{n})$ of diffusion generation mapping $G$ under perturbation $\boldsymbol{n}_x$ along $x$-axis and perturbation $\boldsymbol{n}_y$ along $y$-axis, respectively. The results are shown in (c). The dashed lines show the lower bounds of $\mathcal{A}_G(\boldsymbol{x}, \boldsymbol{n})$ given in Proposition E.1.

proposition will better illustrate this error amplification phenomenon for *non-negligible perturbations*, which we refer to as *instability effect*, when intrinsic instability occurs:

**Proposition 2.1** (Instability effect). *Suppose $F : \mathbb{R}^n \rightarrow \mathbb{R}^n$ is a continuously differentiable mapping. For any $\boldsymbol{x}, \boldsymbol{u} \in \mathbb{R}^n$, and $\Delta > 0$, there exists a non-zero perturbation $\boldsymbol{n} \in \mathbb{R}^n$ such that*

$$\mathcal{A}_F(\boldsymbol{x}, \boldsymbol{n}) := \frac{\|F(\boldsymbol{x} + \boldsymbol{n}) - F(\boldsymbol{x})\|}{\|\boldsymbol{n}\|} \geq \frac{\mathcal{E}_F(\boldsymbol{x}, \boldsymbol{u})}{1 + \Delta}, \tag{3}$$

*where $\mathcal{A}_F(\boldsymbol{x}, \boldsymbol{n})$ is referred to as the instability coefficient. Furthermore, if $\mathcal{E}_F(\boldsymbol{x}, \boldsymbol{u}) > 1$, we have a $\boldsymbol{n}$ that satisfies $\mathcal{A}_F(\boldsymbol{x}, \boldsymbol{n}) > 1$, then we say $F$ exhibits the instability at $\boldsymbol{x}$ under perturbation $\boldsymbol{n}$.*

## 3 INSTABILITY AND ITS AMPLIFICATION ON RECONSTRUCTION ERROR

In this section, we first empirically demonstrate the existence of instability in the diffusion generation mapping $G$ in Section 3.1. Subsequently, we present experimental evidence corroborating the positive correlation between instability and the reconstruction error in Section 3.2.

### 3.1 EMPIRICAL EVIDENCE ON THE EXISTENCE OF INSTABILITY

In Section 2.2, we have defined the intrinsic instability and introduced the instability effect. Here, we will demonstrate that the instability can exist in the generation process of diffusion models.

**Demonstration by a numerical case.** We present a numerical example illustrated in Figure 3. The density function of the diffusion generation distribution is depicted in Figure 3(a). The detailed settings are provided in Section E.1.

In Figure 3(a), the red region is obtained by applying diffusion generation mapping $G$ to the green region. It is evident that $G$ significantly expands the green region along the $y$-axis, indicating that $G$ is highly sensitive to perturbations along the $y$-axis within the green region. Figure 3(b) displays the intrinsic instability coefficient of $G$ along the $y$-axis. It can be observed that the intrinsic instability coefficient along the $y$-axis exceeds one in the central region, indicating that the instability effect can appear in this area. Moreover, this region coincides with the green region in Figure 3(a), demonstrating the relationship between the intrinsic instability and the amplification effect.

We further analyze the instability coefficient $\mathcal{A}_G(\boldsymbol{x}, \boldsymbol{n})$ along the dashed line in Figure 3(b), as well as the lower bound provided by Proposition 2.1. The curves of $\mathcal{A}_G(\boldsymbol{x}, \boldsymbol{n})$ *vs.* $x$ are shown in

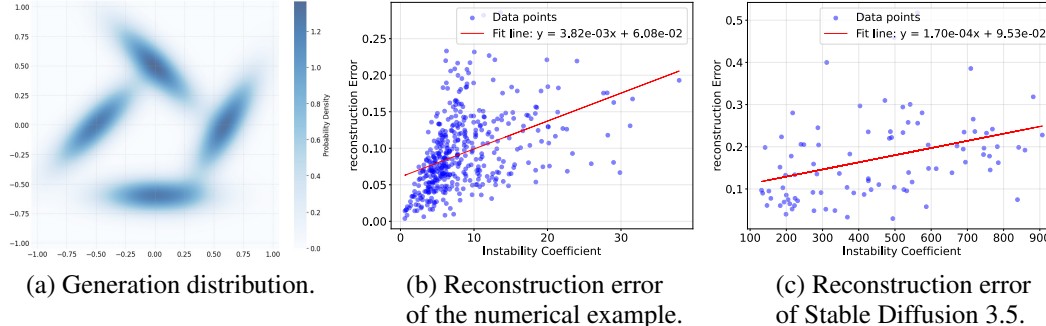

(a) Generation distribution.

(b) Reconstruction error
of the numerical example.

(c) Reconstruction error
of Stable Diffusion 3.5.

Figure 4: **Positive correlation between the reconstruction error and instability coefficient in a numerical case and Stable Diffusion 3.5 Esser et al. (2024).** In numerical experiments, the diffusion model's generation distribution is a two-dimensional mixture of Gaussians. (a) shows the probability density function. (b) displays the relationship between reconstruction error $\mathcal{R}(\boldsymbol{x})$ and instability coefficient. (c) further illustrates the relationship between reconstruction error and instability coefficient in the Stable Diffusion 3.5 model. More results are provided in Section B.1

Figure 3(c). We observe that in the central region, the instability coefficients along $x$ and $y$ axis, *i.e.*, $\mathcal{A}_G(\boldsymbol{x}, \boldsymbol{n}_x)$ and $\mathcal{A}_G(\boldsymbol{x}, \boldsymbol{n}_y)$, are greater than one, further indicating the presence of instability.

## 3.2    INSTABILITY AMPLIFIES RECONSTRUCTION ERROR

This subsection demonstrates that instability in diffusion generation can amplify reconstruction errors. We begin by formally defining the reconstruction error, and provide a theoretical support on the possibility of large reconstruction error. Then we demonstrate the amplification effect by correlation analyses using both numerical cases and practical diffusion models.

**Reconstruction error.** Diffusion reconstruction involves two primary steps for a given data $\boldsymbol{x} \in \mathbb{R}^n$:

1. The diffusion inversion process defined by Definition 2.1 is numerically solved from the given data $\boldsymbol{x}$, which yields an inverted noise $\hat{\boldsymbol{z}} = \widehat{G}^{-1}(\boldsymbol{x})$; and then
2. the diffusion generation process defined by Equation (1) is numerically solved from $\hat{\boldsymbol{z}}$, resulting in a reconstructed data $\hat{\boldsymbol{x}} = \widehat{G}(\hat{\boldsymbol{z}})$,

where $\widehat{G}$ denotes the approximated diffusion generation mapping by numerically solving PF-ODE in Equation (1), and $\widehat{G}^{-1}$ denotes the approximated diffusion inversion mapping, similarly.

The reconstruction error is quantified as the discrepancy between the original data $\boldsymbol{x}$ and the reconstructed data $\hat{\boldsymbol{x}}$. It can be defined as:

$$\mathcal{R}(\boldsymbol{x}) = \frac{1}{\sqrt{n}}\|\boldsymbol{x} - \hat{\boldsymbol{x}}\|_2 = \frac{1}{\sqrt{n}}\|\boldsymbol{x} - \widehat{G}(\hat{\boldsymbol{z}})\|_2 = \frac{1}{\sqrt{n}}\|\boldsymbol{x} - \widehat{G}(\widehat{G}^{-1}(\boldsymbol{x}))\|_2. \tag{4}$$

To analyze the causes of reconstruction error, it is essential to recognize two primary factors:

1. *Discrepancy in inverted noise*: The inverted noise $\hat{\boldsymbol{z}} = \widehat{G}^{-1}(\boldsymbol{x})$ obtained during the diffusion inversion process deviates from the ideal noise $\boldsymbol{z} \sim \mathcal{N}(\boldsymbol{0}, \mathbf{I})$. This discrepancy arises due to numerical discretization errors inherent in solving the PF-ODE Equation (1) and its inverse. As a result, the numerically inverted noise does not perfectly match the ideal one, introducing an initial error into the regeneration pipeline.
2. *Amplification by instability*: The presence of instability in the diffusion generation mapping $\widehat{G}$ can further exacerbate the above discrepancy in the inverted noise, leading to a substantial reconstruction error $\mathcal{R}(\boldsymbol{x})$.

The following proposition further demonstrates that both of the aforementioned two factors contribute to the reconstruction error, even if the diffusion generation mapping $G$ is ideal with infinite numerical precision. A detailed proof is provided in Section D.2.

**Proposition 3.1** (Risk of large reconstruction error). *For any data sample $\boldsymbol{x} \in \mathbb{R}^n$, consider the diffusion reconstruction process consisting of a numerically approximated diffusion inversion under*

*Euler method and a precise diffusion generation process $G$. Let $\hat{z}$ denote the numerically inverted noise, $z$ denote the ideal inverted noise, and $\hat{x}$ denote the reconstructed data. Suppose $\mathcal{E}_G(z, \hat{z} - z) > C$ for some $C > 0$. Then, the upper bound $\mathcal{U}$ of reconstruction error $\mathcal{R}(x)$ satisfies:*

$$\mathcal{U} \geq h M_2 \underbrace{\frac{(C-1)}{2\log C}}_{\substack{\text{Numerical error} \\ \text{in inverted noise}}} \cdot \underbrace{C}_{\substack{\text{Instability} \\ \text{amplification}}} , \tag{5}$$

*where $h$ represents the step size in the numerical solution, and $M_2$ is the estimated upper bound term for the local truncation error of the Euler method.*

**A numerical case.** To verify the amplification effect of instability on reconstruction error, we conduct numerical experiments on a diffusion model with a simple generation distribution. Then, we analyze the correlation between the reconstruction error and the instability coefficient. The density function of the generation distribution is illustrated in Figure 4(a). Detailed settings are provided in Section E.2.

The experimental results are presented in Figure 4(b). We observe a significant correlation between the reconstruction error $\mathcal{R}(x)$ and the instability coefficient. As the instability coefficient increases, the reconstruction error exhibits a noticeable upward trend. This positive correlation corroborates our hypothesis that the instability amplifies reconstruction errors.

**Verification on text-to-image diffusion models.** In practical text-to-image diffusion models, significant reconstruction errors can occasionally occur. To verify that these reconstruction errors are still related to the instability coefficient, we performed a correlation analysis between the reconstruction error $\mathcal{R}(x)$ and the expansion coefficient $\mathcal{E}$ on the Stable Diffusion 3.5 model (Esser et al., 2024). Detailed experimental settings are provided in Section E.2.

The experimental results are illustrated in Figure 4(c). It demonstrate that the reconstruction error $\mathcal{R}(x)$ is significantly positively correlated with the instability coefficient $\mathcal{E}$ in Stable Diffusion 3.5. This finding validates our hypothesis that instability contributes to reconstruction inaccuracies in real-world diffusion models. In Section F, we provide more visual examples of reconstruction failure cases.

## 4 PROBABILISTIC GUARANTEE ON THE OCCURRENCE OF INSTABILITY

In this section, provide theoretical evidence on instability during the ideal diffusion reconstruction process. First, we establish a general probability lower bound without specific assumptions in Section 4.1. For further in-depth analysis, we then discuss the characteristics of the distributions involved in diffusion reconstruction and propose corresponding assumptions in Section 4.2. Finally, based on the distribution assumptions, we provide a theoretical analysis indicating the non-zero probability of instability in Section 4.3.

### 4.1 PROBABILITY LOWER BOUND AS INTUITIVE EVIDENCE FOR INSTABILITY

**Definition of additional instability metric.** Before the formal analysis, we first define *the geometric average of intrinsic instability coefficient* as an indicator of instability.

**Definition 4.1** (Geometric average of intrinsic instability coefficient). Suppose $F : \mathbb{R}^n \to \mathbb{R}^n$ is a continuously differentiable mapping. For any $x \in \mathbb{R}^n$, we define the *geometric average of intrinsic instability coefficient* as

$$\bar{\mathcal{E}}_F(x) := \left( \prod_{i=1}^{n} \mathcal{E}_F(x, u_i) \right)^{\frac{1}{n}}, \tag{6}$$

where $u_1, \cdots, u_n$ is a set of right singular vectors of the Jacobian matrix $J_F(x)$. Note that $\bar{\mathcal{E}}_F(x)$ is also the geometric average of all singular values of $F$'s Jacobian $J_F(x)$.

When $\bar{\mathcal{E}}_F(x) > 1$, at least one of $\mathcal{E}_F(x, u_i)$ is greater than one, thereby indicating the instability.

**Lower bound of instability's probability.** Applying the above instability metric to the diffusion generation mapping $G$, we can now present the following theorem:

**Theorem 4.1** (Probability lower bound of instability). *Suppose $G$ is the ideal diffusion generation mapping defined in Definition 2.1, whose generation distribution is denoted as $\pi_{\text{gen}}$. Let $G^{-1}$ denote its inverse mapping, which represents the ideal diffusion inversion mapping. Further suppose we sample the initial data $\boldsymbol{x}$ from some underlying real distribution $\pi_{\text{real}}$ for reconstruction. Then, for any $M > 0$, we have*

$$\mathcal{P}_M := \pi_{\text{real}}\left(\{\boldsymbol{x} : \bar{\mathcal{E}}_G(G^{-1}(\boldsymbol{x})) > M\}\right) \geq 1 - \epsilon - \delta, \tag{7}$$

$$\epsilon := \pi_{\text{real}}\left(\left\{\boldsymbol{x} : p_{\text{gen}}(\boldsymbol{x}) \geq \frac{1}{(2\pi M^2)^{\frac{n}{2}}} e^{-\frac{2n + 3\sqrt{2n}}{2}}\right\}\right), \tag{8}$$

$$\delta := \pi_{\text{real}}(\{\boldsymbol{x} : \|G^{-1}(\boldsymbol{x})\|^2 > 2n + 3\sqrt{2n}\}), \tag{9}$$

*where $\mathcal{P}_M := \pi_{\text{real}}\left(\{\boldsymbol{x} : \bar{\mathcal{E}}_G(G^{-1}(\boldsymbol{x})) > M\}\right)$ represents the probability of instability in the ideal diffusion reconstruction if $M > 1$, and $p_{\text{gen}}$ denotes the probability density function of $\pi_{\text{gen}}$.*

*More specifically, $\mathcal{P}_M$ describes the probability that the geometric average of intrinsic instability coefficient $\bar{\mathcal{E}}_G(\boldsymbol{z})$ is greater than the threshold $M$ on the inverted noise $\boldsymbol{z} = G^{-1}(\boldsymbol{x})$, where $\boldsymbol{x}$ is sampled from the underlying real distribution $\pi_{\text{real}}$.*

Here, $\epsilon$ denotes the probability that the density function $p_{\text{gen}}(x)$ of $\pi_{\text{gen}}$ at a real data $x$ greater than a threshold $\frac{1}{(2\pi M^2)^{\frac{n}{2}}} e^{-\frac{2n + 3\sqrt{2n}}{2}}$ , while $\delta$ represents the probability that the inverted noise $G^{-1}(\boldsymbol{x})$ of a real data significantly deviates from the center. When both $\epsilon$ and $\delta$ are small enough, and $M > 1$, we can conclude that $\mathcal{P}_M > 0$, and the instability of the diffusion generation mapping is probable to happen during the ideal reconstruction process.

In the next two subsections, we will first make reasonable assumptions based on the characteristics of image data and practical considerations in Section 4.1, and then provide an asymptotic guarantee for the occurrence of instability in Section 4.3.

### 4.2 SETTING DISCUSSIONS FOR IN-DEPTH ANALYSIS

To further analyze the probability $\mathcal{P}_M$ of instability during the diffusion reconstruction process, we first need to make reasonable assumptions about two probability distributions, $\pi_{\text{real}}$ and $\pi_{\text{gen}}$, involved in Theorem 4.1. In the following sections, we will discuss these two distributions separately.

#### 4.2.1 DISCUSSIONS ON THE PROPERTIES OF $\pi_{\text{real}}$.

The distribution $\pi_{\text{real}}$ represents the distribution of images to be reconstructed and is not constrained by the diffusion model itself. We can assume that any real-world image may be used for reconstruction, so we refer to $\pi_{\text{real}}$ as the *real distribution*.

Based on this analysis and the inherent properties of image data, the real distribution $\pi_{\text{real}}$ is characterized by the following two features:

1. **Support as a cube**: The support of $\pi_{\text{real}}$ is a hypercube in $\mathbb{R}^n$. After normalization, it can be assumed that $\text{supp}(\pi_{\text{real}}) = [0, 1]^n$. This is because each pixel in an image has bounded values, typically normalized to the interval $[0, 1]$, ensuring that images in $n$-dimensional space reside within the hypercube $[0, 1]^n$.
2. **Positive minimum density**: The density function $p_{\text{real}}$ of $\pi_{\text{real}}$ has a positive minimum value across its support. This implies that every point within $[0, 1]^n$ corresponds to a possible image, including those that may be uncommon or represent noise-like structures. Although such images have extremely low probability density, they remain potential members of $\pi_{\text{real}}$.

**Assumption 4.1** ($\pi_{\text{real}}$ – Support as a cube). The support of $\pi_{\text{real}}$ is the $n$-dimensional hypercube $[0, 1]^n$, i.e., $\text{supp}(\pi_{\text{real}}) = [0, 1]^n$.

**Assumption 4.2** ($p_{\text{real}}$ – Each pixel gets a chance). The density function $p_{\text{real}}$ of $\pi_{\text{real}}$ is continuous on $\text{supp}(\pi_{\text{real}})$, and $p_{\text{real}}$ further satisfies $\forall \boldsymbol{x} \in [0, 1]^n$, $p_{\text{real}}(\boldsymbol{x}) \geq C_0 > 0$.

#### 4.2.2 DISCUSSIONS ON THE PROPERTIES OF $\pi_{\text{gen}}$.

The generation distribution $\pi_{\text{gen}}$ refers to the distribution of samples generated by the diffusion model when it starts from Gaussian noises. For complex and high-dimensional image data, $\pi_{\text{gen}}$

*typically exhibits sparsity characteristics, meaning it contains several high-probability regions sur-rounded by areas of lower probability.* In theoretical analyses, the mixture of Gaussians is a common choice for modeling such multi-modal distributions. However, in this work, to better differentiate between high and low probability regions, we construct a more general distribution family named as the *mixture of Gaussian neighbors*. The assumption on the density $p_{\text{gen}}$ of $\pi_{\text{gen}}$ is provided here:

**Assumption 4.3** ($p_{\text{gen}}$ – Mixture of Gaussian neighbors). The density function $p_{\text{gen}}$ can be expressed as $p_{\text{gen}} = \sum_{i=1}^{m} a_i f_i * g_{w_i}$, where each $f_i$ is a probability density function supported on the open set $O_i$, $g_{w_i} = \mathcal{N}(0, w_i^2 I)$ is a Gaussian kernel with standard deviation $w_i$, $*$ denotes convolution operator, $m \in \mathbb{Z}^+$, and the coefficients $a_i$ satisfy $a_i > 0$ for all $i = 1, \ldots, m$ and $\sum_{i=1}^{m} a_i = 1$.

In this density function, each density function $f_i$ on open set $O_i$ captures the high density region, while the convolution with the Gaussian kernel $g_{w_i}$ ensures that the surrounding areas have smoothly decreasing probabilities, and thus models the low density region. This approach allows for a flexible representation of complex, multi-modal distributions commonly encountered in high-dimensional image data. The following theorem further demonstrate the approximation capability of the mixture of Gaussian neighbors.

**Theorem 4.2** (Mixture of Gaussian neighbors are Universal Approximators). *The set of density functions $\{p : p = \sum_{i=1}^{m} a_i f_i * g_{w_i}, f_i$ is compactly supported continuous function$\}$ is a dense subset of continuous density function in both $L^2$ metric (Folland, 1999) and Lévy-Prokhorov metric (Billingsley, 1999).*

**Sparsity assumption group.** Combining the model's actual training process, the meaning of the mixture of Gaussian neighbors can be understood as follows: In high-dimensional space for image data, the training samples are sparse and finite due to the limited capability to collect image. When the model is sufficiently trained on the training set and possesses certain generalization capabilities, it can generate a small neighborhood around each training sample with relatively high probability. Thus, each $O_i$ corresponds to the neighborhood that can be generated around each training sample, and $m$ represents the number of training samples. Considering the sparsity of high-dimensional images and the model's limited generalization capabilities, we make the following assumptions:

**Assumption 4.4** (Sparsity assumption group).

1. (Finite training set) We assume that $m$ has an upper bound as the dimension $n$ increases. For convenience, we directly assume that $m$ is a constant.
2. (Sparse data in high-dimensional space) The neighborhoods $O_i$ and $O_j$ do not overlap for any $i \neq j$, and there exists a minimum positive distance $d_{\min} > 0$ between any two distinct neighborhoods $O_i$ and $O_j$. For convenience, we define $\bar{d}_{\min} := d_{\min}/\sqrt{n}$.
3. (Finite generalization capability) Each neighborhood $O_i$ is included in a hypercube $B_i$ of side length $b_i \ll 1$.
4. (Low probability region) For each $f_i$, the probability outside its corresponding $O_i$ is upper bounded by a constant $1 - \alpha_i$, *i.e.*, $\int_{O_i^C} f_i * g_{w_i}(\boldsymbol{x}) \mathrm{d}\boldsymbol{x} < 1 - \alpha_i < 1$.

These assumptions formalize the intuition that in high-dimensional spaces, training samples are finite, sparse, and each training sample can generate a distinct, small neighborhood without overlap. Additionally, the probability density outside these neighborhoods is uniformly bounded, ensuring that low-density regions do not dominate the generation process.

## 4.3 ASYMPTOTIC PROOF OF ALMOST SURE INSTABILITY

Building upon the assumptions on the relevant distributions, in this subsection, we will present a more in-depth asymptotic analysis on $\mathcal{P}_M$, demonstrating that instability occurs almost surely during diffusion reconstruction when the dimensionality $n$ tends to infinity. Regarding real-world high-dimensional images, this asymptotic result implies that instability will occur with a non-negligible probability within the reconstruction process.

Recall that in Theorem 4.1, we have proved that the instability probability $\mathcal{P}_M \geq 1 - \epsilon - \delta$, where $M$ is the threshold of an instability indicator $\bar{\mathcal{E}}_G$ defined in Definition 4.1. To prove a almost sure instability as $n \to \infty$, it is sufficient to prove that $\epsilon \to 0$ and $\delta \to 0$ when $M > 1$. Theorem 4.3 exactly supports this claim. The proof is provided in Section D.5.

**Theorem 4.3.** *Consider $\epsilon$ and $\delta$ defined in Equations (8) and (9) in Theorem 4.1. Suppose that Assumptions 4.1 to 4.3 and Assumption 4.4–the sparsity assumption group–hold. When $n \to \infty$, if $M$ satisfies $M < M_0$, we have*

$$\epsilon := \pi_{\text{real}}\left(\left\{\boldsymbol{x} : p_{\text{gen}}(\boldsymbol{x}) \geq \frac{1}{(2\pi M^2)^{\frac{n}{2}}} e^{-\frac{2n+3\sqrt{2n}}{2}}\right\}\right) \to 0, \tag{10}$$

$$\delta := \pi_{\text{real}}(\{\boldsymbol{x} : \|G^{-1}(\boldsymbol{x})\|^2 > 2n + 3\sqrt{2n}\}) \to 0, \tag{11}$$

*where $M_0 := \min_{1 \leq i \leq m} \exp\left(\frac{1}{8}\frac{\bar{d}_{\min}^2}{w_i} - \ln\frac{1}{w_i} + 2 + 3\sqrt{\frac{2}{n}}\right) \to \infty$. Thus, for any fixed $M > 1$, we can conclude that the instability probability $\mathcal{P}_M \to 1$ as $n \to \infty$.*

The above theorem implies that, for a sufficiently large dimension $n$, such as the case of high-dimensional image data, the instability probability can be high, significantly greater than zero.

**Intuitive explanations.** To better understand Theorem 4.3 and the mechanism underlying this highly-probable instability, we revisit the roles of $\epsilon$ and $\delta$ in the lower bound of the instability probability $\mathcal{P}_M$. The term $\epsilon := \pi_{\text{real}}\left(\left\{\boldsymbol{x} : p_{\text{gen}}(\boldsymbol{x}) \geq \frac{1}{(2\pi M^2)^{\frac{n}{2}}} e^{-\frac{2n+3\sqrt{2n}}{2}}\right\}\right)$ quantifies the probability that samples from $\pi_{\text{real}}$ avoid low probability density regions of $\pi_{\text{gen}}$. As discussed in Section 4.2, the sparsity of $\pi_{\text{gen}}$ ensures that most regions exhibit negligible probability density, thereby driving $\epsilon$ toward zero. Meanwhile, another term $\delta := \pi_{\text{real}}(\{\boldsymbol{x} : \|G^{-1}(\boldsymbol{x})\|^2 > 2n + 3\sqrt{2n}\})$ represents the probability that the inverted noise $G^{-1}(\boldsymbol{x})$ deviates from the Gaussian's center beyond a squared threshold $2n + 3\sqrt{2n}$ — a bound that scales as infinity when dimensionality $n$ increases. It is known that high-dimensional Gaussian samples in $\mathbb{R}^n$ concentrate near a sphere with squared radius $n$. This concentration indicates that significant deviations are of small probability, which implies a small $\delta$. Consequently, both two terms can be small, implying a positive instability probability lower bound $1 - \epsilon - \delta$. A detailed derivation is provided in the Section D.5.

## 5  RELATED WORKS

The concepts of diffusion inversion and reconstruction were initially introduced by Song et al. (2020) as an application of DDIM, and was then used in many image editing tasks (Hertz et al., 2022; Chung et al., 2022a; Su et al., 2022; Tumanyan et al., 2023). However, many studies have identified limitations in their effectiveness for text-to-image diffusion models, particularly in terms of reconstruction quality (Dhariwal & Nichol, 2021; Ho & Salimans, 2022; Zhang et al., 2024; Mokady et al., 2023; Wallace et al., 2023; Hong et al., 2024; Wang et al., 2025; Dai et al., 2024). To improve the reconstruction accuracy, some tuning-based methods are proposed to align the re-generation trajectory to the inversion trajectory(Mokady et al., 2023; Dong et al., 2023). As it is recognized that numerical discretization error is the direct cause of reconstruction inaccuracy, high-order ODE solvers for diffusion models offer alternatives to mitigate this issue (Lu et al., 2022a;b; Karras et al., 2022). Additionally, some works introduce auxiliary variables to ensure the numerical reversibility of the diffusion inversion process (Wallace et al., 2023; Zhang et al., 2024; Wang et al., 2024). However, it has been noted that such methods can be unstable, as small perturbations in the inverted noise may lead to significant deviations in the reconstructed image (Ju et al., 2024). Meanwhile, there remains a lack of comprehensive analysis on the underlying mechanism: why the reconstruction discrepancy can be sufficiently significant. Different from previous works that proposes new methods, this work attempts to dive into the mechanism and advance the understanding of diffusion models.

## 6  CONCLUSION

In this paper, we identify the instability phenomenon as an amplifier on the error observed in diffusion-based image reconstruction. Through rigorous theoretical analysis and comprehensive experimental validations, we demonstrate that instability leads to the amplification of numerical perturbations during the diffusion generation process, thus increasing the reconstruction error. Moreover, we investigate the underlying causes of instability in the diffusion ODE generation process, demonstrating that the inherent sparsity in diffusion generation distribution can cause instability. Meanwhile, we theoretically prove that the instability will almost surely occur during reconstruction for infinite dimensional images. Our work elucidates a critical source of reconstruction inaccuracies. Addressing instability will be essential for advancing the reliability of generative models.

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

CONTENTS OF APPENDIX

## A    PSEUDOCODE FOR DIFFUSION RECONSTRUCTION

---

**Algorithm 1** Diffusion reconstruction

---

    **Input**: Initial data $\boldsymbol{x}$, PF-ODE vector field $\boldsymbol{v}$, ODE solver ODESolver
    **Output**: Reconstructed data $\hat{\boldsymbol{x}}$
    # ODESolver gets four arguments as below
    # ODESolver(initial value, derivative function, initial time, end time)

    # Diffusion inversion from $t = 0$ to $t = 1$
1:  $\hat{\boldsymbol{z}} \leftarrow$ ODESolver$(\boldsymbol{x}, \boldsymbol{v}, 0, 1)$
    # Diffusion regeneration from $t = 1$ to $t = 0$
2:  $\hat{\boldsymbol{x}} \leftarrow$ ODESolver$(\hat{\boldsymbol{z}}, \boldsymbol{v}, 1, 0)$

---

To further illustrate the PF-ODE-based diffusion reconstruction procedure, we provide the pseudocode in Algorithm 1. The pseudocode demonstrates the core logic of the reconstruction porcess, utilizing an ODE solver that could be any numerical method.

## B    ADDITIONAL EXPERIMENTS

### B.1    RECONSTRUCTION ERROR AMPLIFICATION VERIFIED BY SECOND-ORDER ODE SOLVER

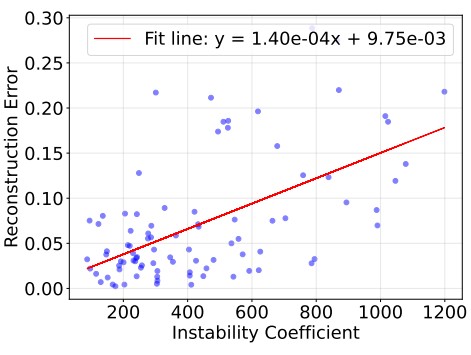

(a) Stable Diffusion 3.5 Medium (Esser et al., 2024)

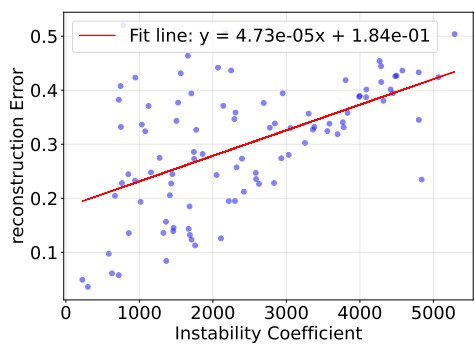

(b) Stable Diffusion 3.5 Large (Esser et al., 2024)

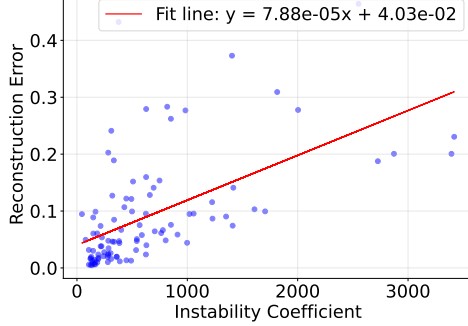

(c) FLUX (Black Forest Labs, 2024)

Figure A1: **Positive correlation between the reconstruction error and instability coefficient verified by second-order Heun ODE solver.** All of the three text-to-image diffusion models exhibit such positive correlation. Experimental details can be found in Section E.

In Section 3.2, we have already demonstrated the positive correlation between the reconstruction error and the instability coefficient of diffusion generation mapping. Both the numerical example

and the results of Stable Diffusion 3.5 Medium (Esser et al., 2024) in Figure 4 imply that the occurrence of instability can amplify the reconstruction error. Note that the Euler solver is adopted in Section 3.2.

Here we further emphasize that **this positive correlation between the reconstruction error and the instability coefficient is an inherent property of PF-ODE, and does not rely on specific numerical method**[2]. To support this, we provide more experimental evidence under the second-order Heun ODE solver, rather than the first-order Euler solver in Section 3.2. The results are illustrated in Figure A1. Here we choose three popular different text-to-image diffusion models– Stable Diffusion 3.5 Medium / Large (Esser et al., 2024) and FLUX (Black Forest Labs, 2024). It can be observed that the positive correlation between the reconstruction error and the instability coefficient again appears in all of these three models.

## C  DISCUSSIONS

**Implication of instability.** Our findings reveal that diffusion generation process can exhibit instability when reconstructing a target data, such as image. Specifically, the reconstructed result may vary dramatically under tiny perturbations on the inverted noise, indicating a high sensitivity that undermines the ability to consistently reproduce the original sample. This behavior sheds light on an important limitation of diffusion models' generalization capability. While diffusion models are often lauded for their strong performance in generating realistic samples, the observed instability implies that they may not have fully learned the underlying structure of the data distribution.

From a theoretical perspective, as discussed in Section 4, such instability can be attributed to the sparsity of the generation distribution: the generation distribution concentrates its probability mass on scattered, small regions in the high-dimensional space, failing to provide robust coverage over the broader data manifold. And this sparsity highlights the model's limited capacity to generalize beyond the concentrated regions. These insights underscore the need for further research into improving how diffusion models capture the global data distribution, mitigate distributional sparsity, and enhance the stability of their generation process.

**Potential solutions** One promising direction to mitigate instability is *distribution regularization*. Since instability fundamentally stems from the sparsity of the learned generation distribution, regularization strategies that encourage more uniform coverage of the latent space may help. For instance, in diffusion models with autoencoder-based latents, additional loss terms or architectural constraints could be introduced to reduce sparsity in the latent distribution, thereby alleviating the instability issue.

## D  DETAILED PROOFS

### D.1  PROOF OF PROPOSITION 2.1

For convenience, we first repeat Proposition 2.1 to be proved here.

**Proposition 2.1** (Instability effect). *Suppose $F : \mathbb{R}^n \to \mathbb{R}^n$ is a continuously differentiable mapping. For any $\boldsymbol{x}, \boldsymbol{u} \in \mathbb{R}^n$, and $\Delta > 0$, there exists a non-zero perturbation $\boldsymbol{n} \in \mathbb{R}^n$ such that*

$$\mathcal{A}_F(\boldsymbol{x}, \boldsymbol{n}) := \frac{\|F(\boldsymbol{x} + \boldsymbol{n}) - F(\boldsymbol{x})\|}{\|\boldsymbol{n}\|} \geq \frac{\mathcal{E}_F(\boldsymbol{x}, \boldsymbol{u})}{1 + \Delta}, \tag{3}$$

*where $\mathcal{A}_F(\boldsymbol{x}, \boldsymbol{n})$ is referred to as the instability coefficient. Furthermore, if $\mathcal{E}_F(\boldsymbol{x}, \boldsymbol{u}) > 1$, we have a $\boldsymbol{n}$ that satisfies $\mathcal{A}_F(\boldsymbol{x}, \boldsymbol{n}) > 1$, then we say $F$ exhibits the instability at $\boldsymbol{x}$ under perturbation $\boldsymbol{n}$.*

*Proof.* Given that the operator norm of the Jacobian matrix $J_F(\boldsymbol{x})$ satisfies $\|J_F(\boldsymbol{x})\| = \sup_{\boldsymbol{u'} \in \mathbb{R}^n} \frac{\|J_F(\boldsymbol{x})\boldsymbol{u'}\|}{\|\boldsymbol{u'}\|} \geq \mathcal{E}_F(\boldsymbol{x}, \boldsymbol{u})$ for any $\boldsymbol{x}, \boldsymbol{u} \in \mathbb{R}^n$, there exists a unit vector $\boldsymbol{h}$ such that

$$\|J_F(\boldsymbol{x})\boldsymbol{h}\| \geq \mathcal{E}_F(\boldsymbol{x}, \boldsymbol{u}). \tag{A1}$$

---

[2]Here "inherent" is with respect to the choice of numerical solver; the phenomenon still depends on the learned generation distribution and its geometry.

Let $\boldsymbol{n}_t = t\boldsymbol{h}$, we have

$$F(\boldsymbol{x} + \boldsymbol{n}_t) - F(\boldsymbol{x}) = J_F(\boldsymbol{x})\boldsymbol{n}_t + R(\boldsymbol{n}_t), \tag{A2}$$

where $R(\boldsymbol{n}_t)$ is the remainder. Then $\mathcal{A}_F(\boldsymbol{x}, \boldsymbol{n}_t)$ is bounded below by:

$$\mathcal{A}_F(\boldsymbol{x}, \boldsymbol{n}_t) := \frac{\|F(\boldsymbol{x} + \boldsymbol{n}_t) - F(\boldsymbol{x})\|}{\|\boldsymbol{n}_t\|} \tag{A3}$$

$$\geq \frac{\|J_F(\boldsymbol{x})\boldsymbol{n}_t\| - \|R(\boldsymbol{n}_t)\|}{\|\boldsymbol{n}_t\|} \tag{A4}$$

$$\geq \frac{t\mathcal{E}_F(\boldsymbol{x}, \boldsymbol{u}) - \|R(\boldsymbol{n}_t)\|}{\|\boldsymbol{n}_t\|} \tag{A5}$$

$$= \mathcal{E}_F(\boldsymbol{x}, \boldsymbol{u}) - \frac{\|R(\boldsymbol{n}_t)\|}{\|\boldsymbol{n}_t\|}, \tag{A6}$$

where the first $\geq$ is by triangle inequality, and the second $\geq$ is by Equation (A1).

Since $F$ is differentiable, the remainder $R(\boldsymbol{n}_t)$ satisfies $\lim_{t \to 0} \frac{\|R(\boldsymbol{n}_t)\|}{\|\boldsymbol{n}_t\|} = 0$. Then for $\epsilon = \frac{\Delta}{1+\Delta}\mathcal{E}_F(\boldsymbol{x}, \boldsymbol{u})$, there exist $\delta > 0$ and $t \leq \delta$ such that $\frac{\|R(\boldsymbol{n}_t)\|}{\|\boldsymbol{n}_t\|} \leq \epsilon$, and this can lead to the conclusion

$$\frac{\|F(\boldsymbol{x} + \boldsymbol{n}_t) - F(\boldsymbol{x})\|}{\|\boldsymbol{n}_t\|} \geq \mathcal{E}_F(\boldsymbol{x}, \boldsymbol{u}) - \frac{\|R(\boldsymbol{n}_t)\|}{\|\boldsymbol{n}_t\|} \geq \frac{\mathcal{E}_F(\boldsymbol{x}, \boldsymbol{u})}{1+\Delta}. \tag{A7}$$

Furthermore, by the arbitrariness of $\Delta$, when $\mathcal{E}_F(\boldsymbol{x}, \boldsymbol{u}) > 1$, there always exists a $\Delta$ satisfies $\mathcal{E}_F(\boldsymbol{x}, \boldsymbol{u}) > 1 + \Delta$. Thus $\mathcal{A}_F(\boldsymbol{x}, \boldsymbol{n}_t) \geq \frac{\mathcal{E}_F(\boldsymbol{x}, \boldsymbol{u})}{1+\Delta} > 1$. $\qquad\square$

### D.2 PROOF OF PROPOSITION 3.1

For convenience, we first repeat Proposition 3.1 to be proved here.

**Proposition 3.1** (Risk of large reconstruction error). *For any data sample $\boldsymbol{x} \in \mathbb{R}^n$, consider the diffusion reconstruction process consisting of a numerically approximated diffusion inversion under Euler method and a precise diffusion generation process $G$. Let $\hat{\boldsymbol{z}}$ denote the numerically inverted noise, $\boldsymbol{z}$ denote the ideal inverted noise, and $\hat{\boldsymbol{x}}$ denote the reconstructed data. Suppose $\mathcal{E}_G(\boldsymbol{z}, \hat{\boldsymbol{z}} - \boldsymbol{z}) > C$ for some $C > 0$. Then, the upper bound $\mathcal{U}$ of reconstruction error $\mathcal{R}(\boldsymbol{x})$ satisfies:*

$$\mathcal{U} \geq \underbrace{hM_2 \frac{(C-1)}{2\log C}}_{\substack{\text{Numerical error} \\ \text{in inverted noise}}} \cdot \underbrace{C}_{\substack{\text{Instability} \\ \text{amplification}}}, \tag{5}$$

*where $h$ represents the step size in the numerical solution, and $M_2$ is the estimated upper bound term for the local truncation error of the Euler method.*

*Proof.* The reconstruction process involves two sequential steps. First, the probability flow ODE in Equation (1) is integrated forward from $t = 0$ to $t = 1$ using the Euler method, yielding the noise corresponding to the image. Then, the same ODE is integrated backward from $t = 1$ to $t = 0$, starting from the noise, to recover the reconstructed image. We use the reconstruction process to evaluate the Lipschitz constant of $\boldsymbol{v}(\boldsymbol{x}, t)$ and to analyze the error between the inversion and reconstruction procedures. The form of the probability flow ODE solved using the Euler method in the inversion process is:

$$\boldsymbol{x}_{t_{n+1}} = \boldsymbol{x}_{t_n} + (t_{n+1} - t_n)\boldsymbol{v}(\boldsymbol{x}_{t_n}, t_n). \tag{A8}$$

The error analysis of Euler's numerical solution involves the estimation of local truncation error (LTE) and global truncation error (GTE), which together determine the method's overall accuracy.

The local truncation error measures the error introduced in a single step of the Euler method. Denoting the time step $t_{n+1} - t_n$ as $h$, by expanding the true solution $\boldsymbol{x}(t)$ at time $t_{n+1}$ using a Taylor series around $t_n$, we obtain:

$$\boldsymbol{x}(t_{n+1}) = \boldsymbol{x}(t_n) + h\boldsymbol{x}'(t_n) + \frac{h^2}{2}\boldsymbol{x}''(\xi), \tag{A9}$$

where $\xi \in [t_n, t_{n+1}]$. Since $\boldsymbol{x}'(t_n) = \boldsymbol{v}(\boldsymbol{x}_{t_n}, t_n)$, the local truncation error is given by assuming $\boldsymbol{x}_{t_n} = \boldsymbol{x}(t_n)$:

$$LTE := \boldsymbol{x}(t_{n+1}) - \boldsymbol{x}_{t_{n+1}} = \frac{h^2}{2}\boldsymbol{x}''(\xi). \tag{A10}$$

If the second derivative of $\boldsymbol{x}$ is bounded, that is, $|\boldsymbol{x}''| \leq M_2$, then the error bound is

$$LTE \leq \frac{1}{2}M_2h^2. \tag{A11}$$

The global truncation error measures the accumulated error over multiple steps, and the total error can be estimated by summing up the contributions from each step. Let $L$ denote the Lipschitz constant of $\boldsymbol{v}$, then the global truncation error $E_n$ satisfies the following recurrence relation:

$$E_{n+1} := \|\boldsymbol{x}_{t_{n+1}} - \boldsymbol{x}(t_{n+1})\| \tag{A12}$$

$$= \|(\boldsymbol{x}_{t_n} + h\boldsymbol{v}(\boldsymbol{x}_{t_n}, t_n) - (\boldsymbol{x}(t_n) + h\boldsymbol{x}'(t_n) + \frac{h^2}{2}\boldsymbol{x}''(\xi))\| \tag{A13}$$

$$\leq \|\boldsymbol{x}_{t_n} - \boldsymbol{x}(t_n)\| + h\|\boldsymbol{v}(\boldsymbol{x}_{t_n}, t_n) - \boldsymbol{v}(\boldsymbol{x}(t_n), t_n)\| + LTE \tag{A14}$$

$$\leq E_n + hL \cdot E_n + \frac{1}{2}M_2h^2. \tag{A15}$$

This yields

$$E_n \leq \frac{hM_2}{2L}\left(e^L - 1\right). \tag{A16}$$

While considering the Gronwall inequality for the precise generation process, we can obtain that for two PF-ODE solutions $\boldsymbol{x}_t$, $\boldsymbol{y}_t$ with distinct initial conditions $\boldsymbol{x}_1$, $\boldsymbol{y}_1$ at time $t = 1$, the exact solutions at time $t = 0$ satisfy the following property:

$$\|\boldsymbol{y}_0 - \boldsymbol{x}_0\| \leq \|\boldsymbol{x}_1 - \boldsymbol{y}_1\|e^L. \tag{A17}$$

This equation will give a upper bound to the intrinsic instability coefficient $\mathcal{E}_G(\boldsymbol{x}, \boldsymbol{u})$:

$$\sup_{\boldsymbol{x},\boldsymbol{u}} \mathcal{E}_G(\boldsymbol{x}, \boldsymbol{u}) \leq e^L. \tag{A18}$$

From Equation (A18) and the assumption $\mathcal{E}_G(\boldsymbol{z}, \hat{\boldsymbol{z}} - \boldsymbol{z}) > C$, we can infer that $L \geq \log C$. Substituting into our previous discussion on the global truncation error, we have:

$$\mathcal{U} \geq \frac{hM_2}{2L}(e^L - 1) \cdot C \geq hM_2\frac{C-1}{2\log C} \cdot C. \tag{A19}$$

$\square$

## D.3 PROOF OF THEOREM 4.1

To prove Theorem 4.1, we first present two lemmas.

**Lemma D.1.** *For any ODE with a unique solution:*

$$\frac{\mathrm{d}\boldsymbol{x}_t}{\mathrm{d}t} = \boldsymbol{v}(\boldsymbol{x}_t, t), \tag{A20}$$

*its solution can be expressed as $\boldsymbol{x}_t = \phi_t(\boldsymbol{x}_0)$ for $t \in [0, 1]$ with the initial condition $\boldsymbol{x}_0$ at $t = 0$. Let $p_t(\boldsymbol{x})$ be a probability density function that satisfies $\int_{\phi_t^{-1}(A)} p_0(\boldsymbol{x})\mathrm{d}\boldsymbol{x} = \int_A p_t(\boldsymbol{x})\mathrm{d}\boldsymbol{x}$ for any measurable set $A$. Then the geometric average of intrinsic instability coefficient for mapping $\phi_1$, i.e., $\bar{\mathcal{E}}_{\phi_1}(\boldsymbol{x})$ defined as Definition 4.1, satisfies:*

$$\left|\bar{\mathcal{E}}_{\phi_1}(\boldsymbol{x})\right|^n \geq \exp\left(\int_0^1 \nabla \cdot \boldsymbol{v}(\boldsymbol{x}_t, t)dt\right) = \frac{p_0(\boldsymbol{x})}{p_1(\phi_1(\boldsymbol{x}))},$$

*Proof.* For any $t \in [0, 1]$ and $\boldsymbol{x}_0$, define $J(t) = \frac{\partial \boldsymbol{x}_t}{\partial \boldsymbol{x}_0}$, where $\boldsymbol{x}_t$ is defined in Equation (A20). Then $J(0) = I$, $J(1) = J_{\phi_1}(\boldsymbol{x}_0)$, and

$$\frac{\mathrm{d}}{\mathrm{d}t}J(t) = \frac{\mathrm{d}}{\mathrm{d}t}\frac{\partial \boldsymbol{x}_t}{\partial \boldsymbol{x}_0} \tag{A21}$$

$$= \frac{\partial \boldsymbol{v}(\boldsymbol{x}_t, t)}{\partial \boldsymbol{x}_0} \tag{A22}$$

$$= \frac{\partial \boldsymbol{v}(\boldsymbol{x}_t, t)}{\partial \boldsymbol{x}_t} \circ J(t). \tag{A23}$$

Multiplying $J^{-1}(t)$ in both sides, we have

$$\frac{\mathrm{d}J(t)}{\mathrm{d}t} \circ J^{-1}(t) = \frac{\partial \boldsymbol{v}(\boldsymbol{x}_t, t)}{\partial \boldsymbol{x}_t}. \tag{A24}$$

Denote the $i$-th eigenvalue of $J(t)$ as $\lambda_i(J(t))$. With Equation (A24), we can obtain $\frac{\mathrm{d}}{\mathrm{d}t} \log |\prod_{i=1}^n \lambda_i(J(t))|$ as follows

$$\frac{\mathrm{d}}{\mathrm{d}t} \log |\prod_{i=1}^n \lambda_i(J(t))| = \sum_{i=1}^n \frac{\mathrm{d}}{\mathrm{d}t} \log |\lambda_i(J(t))| \tag{A25}$$

$$= \sum_{i=1}^n \frac{\mathrm{d}\lambda_i(J(t))}{\mathrm{d}t} \lambda_i(J(t))^{-1} \tag{A26}$$

$$= \mathrm{tr}\left(\frac{\mathrm{d}J(t)}{\mathrm{d}t} \circ J^{-1}(t)\right) \tag{A27}$$

$$= \mathrm{tr}\left(\frac{\partial \boldsymbol{v}(\boldsymbol{x}_t, t)}{\partial \boldsymbol{x}_t}\right) \tag{A28}$$

$$= \nabla \cdot \boldsymbol{v}(\boldsymbol{x}_t, t). \tag{A29}$$

Thus deduce that

$$\log |\prod_{i=1}^n \lambda_i(J(t))| = \int_0^1 \nabla \cdot \boldsymbol{v}(\boldsymbol{x}_t, t)\mathrm{d}t. \tag{A30}$$

On the other hand, for the probability density we can derive that:

$$\frac{\mathrm{d}\log p_t(\boldsymbol{x}_t)}{\mathrm{d}t} = \nabla \log p_t(\boldsymbol{x}_t) \cdot \frac{\mathrm{d}\boldsymbol{x}_t}{\mathrm{d}t} + \frac{\partial}{\partial t} \log p_t(\boldsymbol{x}, t) \tag{A31}$$

$$= \nabla \log p_t(\boldsymbol{x}_t) \cdot \boldsymbol{v}(\boldsymbol{x}_t, t) + \frac{\partial}{\partial t} \log p_t(\boldsymbol{x}, t) \tag{A32}$$

$$= \frac{\nabla p_t(\boldsymbol{x}_t)}{p_t(\boldsymbol{x}_t)} \cdot \boldsymbol{v}(\boldsymbol{x}_t, t) - \frac{\nabla \cdot (p_t(\boldsymbol{x}_t)\boldsymbol{v}(\boldsymbol{x}_t, t))}{p_t(\boldsymbol{x}_t)} \tag{A33}$$

$$= -\nabla \cdot \boldsymbol{v}(\boldsymbol{x}_t, t), \tag{A34}$$

where Equation (A33) is a result from Fokker-Planck equation: $\frac{\partial p_t(\boldsymbol{x}_t)}{\partial t} = -\nabla \cdot (p_t(\boldsymbol{x}_t)\boldsymbol{v}(\boldsymbol{x}_t, t))$. Thus we conclude that

$$\log p_t(\boldsymbol{x}_t) + \log |\prod_{i=1}^n \lambda_i(J(t))| = \log p_0(\boldsymbol{x}_0) + \log |\prod_{i=1}^n \lambda_i(J(0))|. \tag{A35}$$

Note that $J(0)$ is the identity matrix, hence we obtain

$$\left|\bar{\mathcal{E}}_{\phi_1}(\boldsymbol{x})\right|^n \geq |\prod_{i=1}^n \lambda_i(J(t))| = \frac{p_0(\boldsymbol{x}_0)}{p_t(\boldsymbol{x}_t)}, \tag{A36}$$

where the first inequality holds because the product of the singular values of a matrix is greater than the absolute value of the product of its eigenvalues. $\square$

**Lemma D.2.** *Let $\gamma$ be the standard Gaussian probability density function and $G^{-1}$ be the inversion function. For all $\boldsymbol{x} \in \mathrm{supp}(\pi_{\mathrm{data}})$, and for all $K > 0$, $k > 0$, we have*

$$\pi_{\mathrm{real}}\left(\left\{\boldsymbol{x} : \frac{p_{data}(\boldsymbol{x})}{\gamma(G^{-1}(\boldsymbol{x}))} < K\right\}\right) \geq \pi_{\mathrm{real}}\left(\{\boldsymbol{x} : p_{data}(\boldsymbol{x}) < kK\}\right)$$
$$- \pi_{\mathrm{real}}\left(\{\boldsymbol{x} : \gamma(G^{-1}(\boldsymbol{x})) \leq k\}\right). \tag{A37}$$

*Proof.*

$$\pi_{\text{real}} \left( \left\{ \boldsymbol{x} : \frac{p_{data}(\boldsymbol{x})}{\gamma(G^{-1}(\boldsymbol{x}))} < K \right\} \right)$$

$$= \pi_{\text{real}} \left( \{\boldsymbol{x} : p_{data}(\boldsymbol{x}) < K\gamma(G^{-1}(\boldsymbol{x}))\} \cap \{\boldsymbol{x} : \gamma(G^{-1}(\boldsymbol{x})) > k\} \right) \tag{A38}$$

$$+ \pi_{\text{real}} \left( \{\boldsymbol{x} : p_{data}(\boldsymbol{x}) < K\gamma(G^{-1}(\boldsymbol{x}))\} \cap \{\boldsymbol{x} : \gamma(G^{-1}(\boldsymbol{x})) \leq k\} \right) \tag{A39}$$

$$\geq \pi_{\text{real}} \left( \{\boldsymbol{x} : \boldsymbol{x} : p_{data}(\boldsymbol{x}) < K\gamma(G^{-1}(\boldsymbol{x}))\} \cap \{\boldsymbol{x} : \gamma(G^{-1}(\boldsymbol{x})) > k\} \right) \tag{A40}$$

$$\geq \pi_{\text{real}} \left( \{\boldsymbol{x} : \gamma(G^{-1}(\boldsymbol{x})) > k\} \cap \{\boldsymbol{x} : p_{data}(\boldsymbol{x}) < kK\} \right) \tag{A41}$$

$$\geq \pi_{\text{real}} \left( \{\boldsymbol{x} : p_{data}(\boldsymbol{x}) < kK\} \right) - \pi_{\text{real}} \left( \{\boldsymbol{x} : \gamma(G^{-1}(\boldsymbol{x})) \leq k\} \right). \tag{A42}$$

$\square$

Now the conclusion of Theorem 4.1 is a direct corollary of the two lemmas above. For convenience, we first repeat Theorem 4.1 to be proved here.

**Theorem 4.1** (Probability lower bound of instability). *Suppose $G$ is the ideal diffusion generation mapping defined in Definition 2.1, whose generation distribution is denoted as $\pi_{\text{gen}}$. Let $G^{-1}$ denote its inverse mapping, which represents the ideal diffusion inversion mapping. Further suppose we sample the initial data $\boldsymbol{x}$ from some underlying real distribution $\pi_{\text{real}}$ for reconstruction. Then, for any $M > 0$, we have*

$$\mathcal{P}_M := \pi_{\text{real}} \left( \{\boldsymbol{x} : \bar{\mathcal{E}}_G(G^{-1}(\boldsymbol{x})) > M\} \right) \geq 1 - \epsilon - \delta, \tag{7}$$

$$\epsilon := \pi_{\text{real}} \left( \left\{ \boldsymbol{x} : p_{\text{gen}}(\boldsymbol{x}) \geq \frac{1}{(2\pi M^2)^{\frac{n}{2}}} e^{-\frac{2n+3\sqrt{2n}}{2}} \right\} \right), \tag{8}$$

$$\delta := \pi_{\text{real}}(\{\boldsymbol{x} : \|G^{-1}(\boldsymbol{x})\|^2 > 2n + 3\sqrt{2n}\}), \tag{9}$$

*where $\mathcal{P}_M := \pi_{\text{real}} \left( \{\boldsymbol{x} : \bar{\mathcal{E}}_G(G^{-1}(\boldsymbol{x})) > M\} \right)$ represents the probability of instability in the ideal diffusion reconstruction if $M > 1$, and $p_{\text{gen}}$ denotes the probability density function of $\pi_{\text{gen}}$.*

*More specifically, $\mathcal{P}_M$ describes the probability that the geometric average of intrinsic instability coefficient $\bar{\mathcal{E}}_G(\boldsymbol{z})$ is greater than the threshold $M$ on the inverted noise $\boldsymbol{z} = G^{-1}(\boldsymbol{x})$, where $\boldsymbol{x}$ is sampled from the underlying real distribution $\pi_{\text{real}}$.*

*Proof.* From Lemma D.1 and Lemma D.2, we have

$$\mathcal{P}_M = \pi_{\text{real}} \left( \{\boldsymbol{x} : \bar{\mathcal{E}}_G(G^{-1}(\boldsymbol{x})) > M\} \right) \tag{A43}$$

$$\geq \pi_{\text{real}} \left( \left\{ \boldsymbol{x} : \frac{p_{\text{gen}}(\boldsymbol{x})}{\gamma(G^{-1}(\boldsymbol{x}))} < M^{-n} \right\} \right) \tag{A44}$$

$$\geq \pi_{\text{real}} \left( \left\{ \boldsymbol{x} : p_{\text{gen}}(\boldsymbol{x}) < \frac{1}{(2\pi M^2)^{\frac{n}{2}}} e^{-\frac{2n+3\sqrt{2n}}{2}} \right\} \right)$$

$$- \pi_{\text{real}} \left( \left\{ \boldsymbol{x} : \gamma(G^{-1}(\boldsymbol{x})) < \frac{1}{(2\pi)^{\frac{n}{2}}} e^{-\frac{2n+3\sqrt{2n}}{2}} \right\} \right), \tag{A45}$$

where the first inequality Equation (A44) follows from Lemma D.1 if we take $p_0$ as the Gaussian density function $\gamma$ and $p_t$ as the density of generated distribution $p_{\text{gen}}$ during the generation process, the second inequality Equation (A45) follows from Lemma D.2 with $k = \frac{1}{(2\pi)^{\frac{n}{2}}} e^{-\frac{2n+3\sqrt{2n}}{2}}$. $\square$

### D.4 PROOF OF THEOREM 4.2

Before the formal proof of Theorem 4.2, we first provide the definition of Lévy-Prokhorov metric (Billingsley, 1999).

#### D.4.1 DEFINITION OF LÉVY-PROKHOROV METRIC

**Definition D.1** (Lévy-Prokhorov metric). For a subset $A \subset \mathbb{R}^n$, define the $\epsilon$- neighborhood of $A$ by

$$A^\epsilon := \{p \in \mathbb{R}^n : \exists q \in A, \ d(p,q) < \epsilon\} = \bigcup_{p \in A} B_\epsilon(p). \tag{A46}$$

where $B_\epsilon(p)$ is the open ball of radius $\epsilon$ centered at $p$. Let $\mathcal{P}(\mathbb{R}^n)$ denotes the collection of all probability measures on $\mathbb{R}^n$ The Lévy-Prokhorov metric $\pi : \mathcal{P}(\mathbb{R}^n)^2 \to [0, +\infty)$is defined as follows:

$$\pi(\mu, \nu) := \inf\{\epsilon > 0 : \mu(A) \le \nu(A^\epsilon) + \epsilon \text{ and } \nu(A) \le \mu(A^\epsilon) + \epsilon\}. \quad (A47)$$

This metric is equivalent to weak convergence of measures (Billingsley, 1999).

### D.4.2 PROOF OF THEOREM 4.2

For convenience, we first repeat Theorem 4.2 to be proved here.

**Theorem 4.2** (Mixture of Gaussian neighbors are Universal Approximators). *The set of density functions $\{p : p = \sum_{i=1}^m a_i f_i * g_{w_i}, f_i \text{ is compactly supported continuous function}\}$ is a dense subset of continuous density function in both $L^2$ metric (Folland, 1999) and Lévy-Prokhorov metric (Billingsley, 1999).*

*Proof.* $n$-dimensional Hermite functions form a complete orthogonal basis of $L^2(\mathbb{R}^n)$ space (Stein & Shakarchi, 2003). Hermite functions can be expressed as:

$$\phi_\alpha(\boldsymbol{x}) = H_\alpha(\boldsymbol{x})e^{-\|\boldsymbol{x}\|^2}. \quad (A48)$$

Here $\alpha$ is a multi-index, an ordered $n$-tuple of nonnegative integers. $H_\alpha(\boldsymbol{x})$ is a polynomial of $\alpha$ order called Hermite polynomials. By Plancherel theorem, the Fourier transform is a isomorphism on $L^2(\mathbb{R}^n)$. (Folland, 2013) It is sufficient to prove the Fourier transform of $\{p(\boldsymbol{x}) = \sum_{i=1}^m a_i f_i * g_{w_i}(\boldsymbol{x})\}$ can converge to Hermite functions in $L^2$. We denote the Fourier transform of any function $h$ as $\hat{h}$, with the wide hat notation $\widehat{\cdot}$. Thus

$$\sum_{i=1}^m a_i \widehat{f_i * g_{w_i}}(\boldsymbol{x}) = \sum_{i=1}^m a_i \hat{f}_i \cdot e^{-\frac{1}{2}w_i^2\|\boldsymbol{x}\|^2}. \quad (A49)$$

Since the continuous compact support function $C_c(\mathbb{R}^n)$ is dense in $L^2(\mathbb{R}^n)$ (Folland, 2013), by the isomorphism, $\{a_i\hat{f}_i\}$ is also dense in $L^2(\mathbb{R}^n)$. $H_\alpha(\boldsymbol{x})e^{-\frac{1}{2}\|\boldsymbol{x}\|^2}$ is a $L^2$ function. Then we derive that $\forall \phi_\alpha, \forall \delta > 0, \exists af$ and take $w_i = 1$,

$$\|\widehat{af * g_1}(\boldsymbol{x}) - \phi_\alpha(\boldsymbol{x})\|_{L^2} = \|(H_\alpha(\boldsymbol{x})e^{-\frac{1}{2}\|\boldsymbol{x}\|^2} - a\hat{f})e^{-\frac{1}{2}\|\boldsymbol{x}\|^2}\|_{L^2} \quad (A50)$$

$$\le \|(H_\alpha(w_i\boldsymbol{x})e^{-\frac{1}{2}\|\boldsymbol{x}\|^2} - a\hat{f})\|_{L^2}\|e^{-\frac{1}{2}\|\boldsymbol{x}\|^2}\|_{L^\infty} \quad (A51)$$

$$\le \delta. \quad (A52)$$

The first inequality is Hölder's inequality, and the second inequality follows from the density of $af$ in the $L^2$ space. $\forall \epsilon > 0, h(x) \in L^2(\mathbb{R}^n)$, there exists a finite set of multi-index $\{\alpha_k, k \le N\}$ and sequence $\{\theta_{\alpha_k}\}$ such that

$$\|\sum_{k \le N} \theta_{\alpha_k}\phi_{\alpha_k}(\boldsymbol{x}) - \hat{h}(\boldsymbol{x})\|_{L^2} \le \frac{\epsilon}{2}, \quad (A53)$$

then for each $\phi_{\alpha_k}$ choose a $a_{\alpha_k}f_{\alpha_k} * g_{w_{\alpha_k}}$ such that $\|a_{\alpha_k}f_{\alpha_k} * g_{w_{\alpha_k}}(\boldsymbol{x}) - \theta_{\alpha_k}\phi_{\alpha_k}(\boldsymbol{x})\|_{L^2} \le \frac{\epsilon}{2N}$, we obtain that

$$\|\sum_{k \le N} a_{\alpha_k}f_{\alpha_k} * g_{w_{\alpha_k}}(\boldsymbol{x}) - \hat{h}(\boldsymbol{x})\|_{L^2} \quad (A54)$$

$$\le \|\sum_{k \le N} a_{\alpha_k}f_{\alpha_k} * g_{w_{\alpha_k}}(\boldsymbol{x}) - \sum_{k \le N} \theta_{\alpha_k}\phi_{\alpha_k}(\boldsymbol{x})\|_{L^2} + \|\sum_{k \le N} \theta_{\alpha_k}\phi_{\alpha_k}(\boldsymbol{x}) - \hat{h}(\boldsymbol{x})\|_{L^2} \quad (A55)$$

$$\le \epsilon, \quad (A56)$$

which shows the density of $\sum_{k \le N} a_i f_i * g_{w_i}$ in $L^2(\mathbb{R})$. Now we prove the density in Lévy-Prokhorov metric, $\forall f$ is a density function of $\mathbb{R}^n$, $\forall \epsilon > 0$, we can construct an $\epsilon$-decomposition as follow: Since $f$ is integrable, there is a compact set $K_f$ in $\mathbb{R}$ with $\int_{K_f^c} f(x)\mathrm{d}\boldsymbol{x} < \epsilon$ and $\int_{\partial K_f} f(\boldsymbol{x})\mathrm{d}\boldsymbol{x} = 0$. The interior of K is an open set $O$ and by the structure theorem of open sets, there exists a decomposition $O = \bigcup_i O_i$, $\{O_i\}$ are disjoint open sets and to make the index set finite, the distance between $O_i$ is no less than some $d > 0$. Then we have $O = \bigcup_{i \le N'} O_i$

Then we define $a_i := \int_{O_i} f(\boldsymbol{x}) \mathrm{d}\boldsymbol{x}$ and $f_i(\boldsymbol{x}) := \frac{1}{a_i} f(\boldsymbol{x}) I_{O_i}$, $\forall A' \subset \bigcup_{i \leq N'} O_i$, denote $A' \cap O_j$ as $A'_j$

$$\int_{A'} |f - \sum_i a_i f_i * g_{w_i}| \mathrm{d}\boldsymbol{x} \tag{A57}$$

$$= \sum_j \int_{A'_j} |f - \sum_i a_i f_i * g_{w_i}| \mathrm{d}\boldsymbol{x} \tag{A58}$$

$$\leq \sum_j \int_{A'_j} |f - a_j f_j * g_{w_j}| \mathrm{d}\boldsymbol{x} + \sum_j \int_{A'_j} |a_j f_j * g_{w_j} - \sum_i a_i f_i * g_{w_i}| \mathrm{d}\boldsymbol{x}. \tag{A59}$$

$g_{w_j}$ is an approximate identity about $w_j$ and when $w_j$ is sufficiently small, $\int_{A'_j} |f - a_j f_j * g_{w_j}| \mathrm{d}\boldsymbol{x} \leq \|f - a_j f_j * g_{w_j}\|_{L^1} \leq \frac{\epsilon}{2N'}$ and

$$\int_{A'_i} |a_j f_j * g_{w_j} - \sum_i a_i f_i * g_{w_i}| \mathrm{d}\boldsymbol{x} = \sum_{i \neq j} \int_{A'_i} |a_i f_i * g_{w_i}| \mathrm{d}\boldsymbol{x} \leq \sum_{i \neq j} \frac{1}{2\pi w_i} \exp(-\frac{d^2}{2w_i^2}), \tag{A60}$$

when $w_i$ is sufficiently small, $\sum_{i \neq j} \frac{1}{2\pi w_i} \exp(-\frac{d^2}{2w_i^2}) \leq \frac{\epsilon}{2N'}$. From this we conclude that $\forall A' \subset \bigcup_{i \leq N'} O_i$, we have

$$\int_{A'} f(\boldsymbol{x}) \mathrm{d}\boldsymbol{x} \leq \int_{A'} \sum_i a_i f_i * g_{w_i}(\boldsymbol{x}) \mathrm{d}\boldsymbol{x} + \epsilon, \tag{A61}$$

$$\int_{A'} \sum_i a_i f_i * g_{w_i}(\boldsymbol{x}) \mathrm{d}\boldsymbol{x} \leq \int_{A'} f(\boldsymbol{x}) \mathrm{d}\boldsymbol{x} + \epsilon. \tag{A62}$$

And $\forall A' \subset O^c$, $\int_{A'} f(\boldsymbol{x}) \mathrm{d}\boldsymbol{x} \leq \epsilon$, then it is sufficient to prove

$$\int_{A'} \sum_i a_i f_i * g_{w_i}(\boldsymbol{x}) \mathrm{d}\boldsymbol{x} \leq \int_{A'^\epsilon} f(\boldsymbol{x}) \mathrm{d}\boldsymbol{x} + \epsilon \tag{A63}$$

and this could also be obtained by the $L^1$ convergence property of approximate identity: $\int_{A'} \sum_i a_i f_i * g_{w_i}(\boldsymbol{x}) \mathrm{d}\boldsymbol{x} < \epsilon$ with appropriate $w_i$.

In summary, $\forall A' \subset \mathbb{R}^n$,

$$\int_{A'} f(\boldsymbol{x}) \mathrm{d}\boldsymbol{x} = \int_{A' \cap O} f(\boldsymbol{x}) \mathrm{d}\boldsymbol{x} + \int_{A' \cap O^c} f(\boldsymbol{x}) \mathrm{d}\boldsymbol{x} \tag{A64}$$

$$\leq \int_{(A' \cap O)^\epsilon} \sum_i a_i f_i * g_{w_i}(\boldsymbol{x}) \mathrm{d}\boldsymbol{x} + \epsilon + \epsilon \tag{A65}$$

$$\leq \int_{(A')^{2\epsilon}} \sum_i a_i f_i * g_{w_i}(\boldsymbol{x}) \mathrm{d}\boldsymbol{x} + 2\epsilon, \tag{A66}$$

$$\int_{A'} \sum_i a_i f_i * g_{w_i}(\boldsymbol{x}) \mathrm{d}\boldsymbol{x} = \int_{A' \cap O} \sum_i a_i f_i * g_{w_i}(\boldsymbol{x}) \mathrm{d}\boldsymbol{x} + \int_{A' \cap O^c} \sum_i a_i f_i * g_{w_i}(\boldsymbol{x}) \mathrm{d}\boldsymbol{x} \tag{A67}$$

$$\leq \int_{(A' \cap O)^\epsilon} f(\boldsymbol{x}) \mathrm{d}\boldsymbol{x} + \epsilon + \epsilon \tag{A68}$$

$$\leq \int_{(A')^{2\epsilon}} f(\boldsymbol{x}) \mathrm{d}\boldsymbol{x} + 2\epsilon. \tag{A69}$$

By the arbitrariness of $\epsilon$, we conclude that the set of distributions with probability density function $\{\sum_i a_i f_i * g_{w_i}\}$ is dense. $\qquad\square$

### D.5 PROOF OF THEOREM 4.3

For convenience, we first repeat Theorem 4.3 to be proved here.

**Theorem 4.3.** *Consider $\epsilon$ and $\delta$ defined in Equations* (8) *and* (9) *in Theorem 4.1. Suppose that Assumptions 4.1 to 4.3 and Assumption 4.4–the sparsity assumption group–hold. When $n \to \infty$, if $M$ satisfies $M < M_0$, we have*

$$\epsilon := \pi_{\text{real}} \left( \left\{ \boldsymbol{x} : p_{\text{gen}}(\boldsymbol{x}) \geq \frac{1}{(2\pi M^2)^{\frac{n}{2}}} e^{-\frac{2n+3\sqrt{2n}}{2}} \right\} \right) \to 0, \tag{10}$$

$$\delta := \pi_{\text{real}}(\{ \boldsymbol{x} : \|G^{-1}(\boldsymbol{x})\|^2 > 2n + 3\sqrt{2n} \}) \to 0, \tag{11}$$

*where $M_0 := \min_{1 \leq i \leq m} \exp\left( \frac{1}{8} \frac{\bar{d}_{\min}^2}{w_i} - \ln \frac{1}{w_i} + 2 + 3\sqrt{\frac{2}{n}} \right) \to \infty$. Thus, for any fixed $M > 1$, we can conclude that the instability probability $\mathcal{P}_M \to 1$ as $n \to \infty$.*

*Proof.* The overall idea of the proof is to find a value $M$ related to $w_i$ such that as $n \to \infty$, we have $w_i \to 0$, thereby implying that both $\epsilon$ and $\delta$ tend to 0 and $M \to \infty$. From Assumption 4.3

$$p_{gen}(\boldsymbol{x}) = \sum_{i=1}^{m} \int_{\mathbb{R}^n} a_i f_i(\boldsymbol{y}) g_{w_i}(\boldsymbol{x} - \boldsymbol{y}) \mathrm{d}\boldsymbol{y} \tag{A70}$$

$$= \sum_{i=1}^{m} \int_{\underset{i \leq m}{\bigcup} O_i} a_i f_i(\boldsymbol{y}) \frac{1}{(2\pi w_i^2)^{\frac{n}{2}}} e^{-\frac{\|\boldsymbol{x}-\boldsymbol{y}\|^2}{2w_i^2}} \mathrm{d}\boldsymbol{y} \tag{A71}$$

$$\leq \sum_{i=1}^{m} \int_{\underset{i \leq m}{\bigcup} O_i} a_i f_i(y) \frac{1}{(2\pi w_i^2)^{\frac{n}{2}}} e^{-\frac{\min_{i \leq m} d(\boldsymbol{x},O_i)^2}{2w_i^2}} \mathrm{d}\boldsymbol{y} \tag{A72}$$

$$= \sum_{i=1}^{m} a_i \frac{1}{(2\pi w_i^2)^{\frac{n}{2}}} e^{-\frac{\min_{i \leq m} d(\boldsymbol{x},O_i)^2}{2w_i^2}} \tag{A73}$$

$$\leq \max_{i \leq m} \left\{ \frac{1}{(2\pi w_i^2)^{\frac{n}{2}}} e^{-\frac{\min_{i \leq m} d(\boldsymbol{x},O_i)^2}{2w_i^2}} \right\}. \tag{A74}$$

We define a dominating function

$$h_{w_1,\dots,w_m}(\boldsymbol{x}) := \max_{i \leq m} \left\{ \frac{1}{(2\pi w_i^2)^{\frac{n}{2}}} e^{-\frac{\min_{i \leq m} d(\boldsymbol{x},O_i)^2}{2w_i^2}} \right\}, \tag{A75}$$

which is monotonically decreasing about $\min_{i \leq m} d(\boldsymbol{x}, O_i)$ and can be written as $h_{w_1,\dots,w_m}(x) = h'_{w_1,\dots,w_m}(\min_{i \leq m} d(\boldsymbol{x}, O_i))$, where $h'_{w_1,\dots,w_m}(x) := \max_{i \leq m} \left\{ \frac{1}{(2\pi w_i^2)^{\frac{n}{2}}} e^{-\frac{x^2}{2w_i^2}} \right\}$. From the above inequality we obtain:

$$\left\{ \boldsymbol{x} : p_{\text{gen}}(\boldsymbol{x}) \geq \frac{1}{(2\pi M^2)^{\frac{n}{2}}} e^{-\frac{2n+3\sqrt{2n}}{2}} \right\} \tag{A76}$$

$$\subset \left\{ \boldsymbol{x} : h_{w_i,\dots,w_m}(\boldsymbol{x}) \geq \frac{1}{(2\pi M^2)^{\frac{n}{2}}} e^{-\frac{2n+3\sqrt{2n}}{2}} \right\} \tag{A77}$$

$$= \{ \boldsymbol{x} : \min_{i \leq m} d(\boldsymbol{x}, O_i) \leq r_M \}, \tag{A78}$$

where

$$r_M := \| h'^{-1}_{w_i,\dots,w_m} \left( \frac{1}{(2\pi M^2)^{\frac{n}{2}}} e^{-\frac{2n+3\sqrt{2n}}{2}} \right) \|. \tag{A79}$$

Based on this, taking $M \leq M_0 := \min_{1 \leq i \leq m} \exp\left( \frac{1}{8} \frac{\bar{d}_{\min}^2}{w_i} - \ln \frac{1}{w_i} + 2 + 3\sqrt{\frac{2}{n}} \right)$, we observe that $M_0 \to \infty$ as $w_i \to 0$, and in the following part we will proof $w_i \to 0$ as $n \to \infty$. Here $\bar{d}_{\min}$ is the minimum of distance between $B_i$.

Since $h'_{w_1,\dots,w_m}$ is radial, once $M$ is fixed, $r_M$ is well defined. Thus there exists a unique $w_j$ in Equation (A75) such that for any $\boldsymbol{x}$ satisfying $\min_{i \leq m} d(\boldsymbol{x}, O_i) = r_M$, we have

$$h_{w_1,\dots,w_m}(\boldsymbol{x}) = \frac{1}{(2\pi w_j^2)^{\frac{n}{2}}} e^{-\frac{\min_{i \leq m} d(\boldsymbol{x},O_i)^2}{2w_j^2}}. \tag{A80}$$

Then from Equation (A78) we deduce that

$$\left\{ \boldsymbol{x} : p_{\text{gen}}(\boldsymbol{x}) \geq \frac{1}{(2\pi M^2)^{\frac{n}{2}}} e^{-\frac{2n+3\sqrt{2n}}{2}} \right\} \tag{A81}$$

$$\subset \left\{ \min_{i \leq m} d(\boldsymbol{x}, O_i) \leq \frac{1}{2}\sqrt{nw_j}\bar{d}_{\min} \right\} = \bigcup_{i \leq m} \left\{ d(\boldsymbol{x}, O_i) \leq \frac{1}{2}\sqrt{nw_j}\bar{d}_{\min} \right\}, \tag{A82}$$

where $\bar{d}_{\min}$ is the dimension-normalized $d_{\min}$, as defined in Assumption 4.4. As we discussed previously in the third of Assumption 4.4, all $O_i$ is contained by a cube $B_i$, consequently $\left\{ d(\boldsymbol{x}, O_i) \leq \frac{1}{2}\sqrt{nw_j}\bar{d}_{\min} \right\} \subset \left\{ d(\boldsymbol{x}, B_i) \leq \frac{1}{2}\sqrt{nw_j}\bar{d}_{\min} \right\}$ and to calculate the Lebesgue measure of the geometric body obtained by expanding an $n$-dimensional cube $B_i$ outward by a distance $\frac{1}{2}\sqrt{nw_j}\bar{d}_{\min}$, we can utilize the Steiner formula (Schneider, 1993), which expresses the volume as the sum of the expansion volumes of different-dimensional faces of the original cube, with each expanded volume contribution being the product of the corresponding ball volume.

For each $k \leq n$, the number of $k$-dimensional faces is $\binom{n}{k} \cdot 2^{n-k}$ and each has a expansion volume contribution $b_i^k \cdot \omega_{n-k}(\frac{1}{2}\sqrt{nw_j}\bar{d}_{\min})^{n-k}(\frac{1}{2})^{n-k}$. Based on this, we conclude that

$$m\left\{ d(\boldsymbol{x}, B_i) \leq \frac{1}{2}\sqrt{nw_j}\bar{d}_{\min} \right\} = \sum_{k=0}^{n} \binom{n}{k} b_i^k \cdot \omega_{n-k}(\frac{1}{2}\sqrt{nw_j}\bar{d}_{\min})^{n-k}. \tag{A83}$$

Comparing with the support of $\pi_{\text{real}}$, which contain some disjoint cubes centered at $x_i$ and the edge length is no less than $\sqrt{3\pi e}b_i$, we can make the following analysis:

$$\frac{m\left\{ d(\boldsymbol{x}, O_i) \leq \frac{1}{2}\sqrt{nw_j}\bar{d}_{\min} \right\}}{(\sqrt{3\pi e}b_i)^n} \leq \frac{\pi^{\frac{n}{2}}(b_i\sqrt{n} + \frac{1}{2}\sqrt{nw_j}\bar{d}_{\min})^n}{\Gamma(\frac{n}{2}+1)(\sqrt{3\pi e}b_i)^n} \tag{A84}$$

$$\sim \frac{\pi^{\frac{n}{2}}}{\sqrt{\pi n}(\frac{n}{2e})^{\frac{n}{2}}} \frac{(b_i\sqrt{n} + \frac{1}{2}\sqrt{nw_j}\bar{d}_{\min})^n}{(\sqrt{3\pi e}b_i)^n} \quad (\textit{Stirling formula}). \tag{A85}$$

With the low probability region assumption in Assumption 4.4, we have

$$\int_{O_i} f_i * g_{w_i}(\boldsymbol{x})\mathrm{d}\boldsymbol{x} \tag{A86}$$

$$\leq \int_{B_i} f_i * g_{w_i}(\boldsymbol{x})\mathrm{d}\boldsymbol{x} \tag{A87}$$

$$= \int_{B_i} \int f_i(\boldsymbol{y})g_{w_i}(\boldsymbol{x} - \boldsymbol{y})\mathrm{d}\boldsymbol{y}\mathrm{d}\boldsymbol{x} \tag{A88}$$

$$= \int f_i(\boldsymbol{y}) \int_{B_i} g_{w_i}(\boldsymbol{x} - \boldsymbol{y})\mathrm{d}\boldsymbol{x}\mathrm{d}\boldsymbol{y} \tag{A89}$$

$$\leq \int f_i(\boldsymbol{y}) \int_{B_i} g_{w_i}(\boldsymbol{x} - \boldsymbol{x}_i)\mathrm{d}\boldsymbol{x}\mathrm{d}\boldsymbol{y} \tag{A90}$$

$$= \int_{B_i} g_{w_i}(\boldsymbol{x} - \boldsymbol{x}_i)\mathrm{d}\boldsymbol{x} \tag{A91}$$

$$\leq \left( \int_{-b_i}^{b_i} \frac{1}{2\pi w_i^2} e^{-\frac{x^2}{2w_i^2}} \, dx \right)^n. \tag{A92}$$

As mentioned previously we obtain that $\int_{O_i} f_i * g_{w_i}(\boldsymbol{x})\mathrm{d}\boldsymbol{x} > \alpha_i$ leads to $w_i$ converging to zero when $n \to \infty$, which means $M_0 \to \infty$. Substituting this result into Equation (A85), we obtain: $\frac{m\left\{ d(\boldsymbol{x}, O_i) \leq \frac{1}{2}\sqrt{nw_j}\bar{d}_{\min} \right\}}{(\sqrt{3\pi e}b_i)^n}$ converge to 0 as $n \to \infty$. Thus we can deduce that

$$\lim_{n \to \infty} \frac{m\left\{ \bigcup_{i \leq m} \left\{ d(\boldsymbol{x}, O_i) \leq \frac{1}{2}\sqrt{nw_j}\bar{d}_{\min} \right\} \right\}}{m\left\{ \text{supp}(\pi_{\text{real}}) \right\}} = 0. \tag{A93}$$

See $\pi_{\text{real}}$ on $\mathbb{R}^n$ as taking finite pixels from a continuous function $f \in C(S)$ representing a infinitely precise real image. $S$ is a compact subset of $\mathbb{R}^2$. The pixels form a finite $\epsilon$-dense set $S_n$ of the separable space $S$. And $S_\infty = \bigcup_{n=1}^\infty S_n$ is a countable dense set of S. And all the continuous function on $S_\infty$ is $C(S)$. $\bigcup_n \mathcal{B}(\mathbb{R}^n)$ forms a semialgebra with a premeasure that generates $\mathcal{B}(C(S))$. With the Kolmogorov extension theorem (Tao, 2021), there is a unique distribution $\pi_{\text{real}}'$ on $\mathcal{B}(C(S))$ generated by all $\pi_{\text{real}}^{(n)}$ on $\mathbb{R}^n$. Additionally, considering that step functions could uniformly converge to any $f \in C(S)$. The finite n-pixel image space can also be seen as step functions on $S$: $S = \bigcup_{i \leq n} U_i$ is a decomposition of disjoint set, each $U_i$ represent a pixel and $m(U_i) = \frac{m(S)}{n}$. For any $C(S)$ value random variable X, there is a step function value random variable $X_n = \sum_{i \leq n} \boldsymbol{x}_i I_{U_i}$ as $X$'s n-dimensional projection and $X_n$ converges to $X$ almost surely in the function space equipped with the uniform norm:

$$P(\omega : \|X_n(\omega) - X(\omega)\| \to 0) = 1. \tag{A94}$$

From this, we derive the convergence of probability law :

$$P^{X_n} \xrightarrow{w} P^X, \tag{A95}$$

which means $\pi_{\text{real}}^{(n)}$ converge to $\pi_{\text{real}}'$ in the Banach space of continuous and step functions on S with uniform norm (Ikeda & Watanabe, 2014).

Since $\pi_{\text{real}}^{(n)}$ is absolutely continuous about $\pi_{gen}$ till the infinite-dimensional case, $\delta' = \pi_{gen}\left(G(\boldsymbol{z}) : \|z\|^2 \geq 2n + 3\sqrt{2n}\right) = 1 - F(2n + 3\sqrt{2n}; n)$, where $F(\cdot; n)$ is the cumulative distribution function of chi-square distribution $\chi_n^2$ and $1 - F(2n + 3\sqrt{2n}; n)$ converges to 0. And the set $B_n = \{\|z\|^2 \geq 2n + 3\sqrt{2n}\}$ when placed in the function space means $\|f_z\|_{L^2} \geq m(S)(2 + 3\sqrt{\frac{2}{n}})$, $f_z$ is the step function associated with $z$, as a result we obtain $B_n \subset B_{n+1}$, so $B_\infty = \lim_{n \to \infty} B_n = \bigcup_{n=1}^\infty B_n$ and $\lim_{n \to \infty} \pi_{\text{real}}^{(n)}(B_n) = \pi_{\text{real}}'(B_\infty)$. While $\pi'_{gen}(B_\infty) = \lim_{n \to \infty} \pi_{gen}\left(G(\boldsymbol{z}) : \|z\|^2 \geq 2n + 3\sqrt{2n}\right) = 0$, we derive that $\lim_{n \to \infty} \delta = \lim_{n \to \infty} \pi_{\text{real}}\left(G(\boldsymbol{z}) : \|z\|^2 \geq 2n + 3\sqrt{2n}\right) = \lim_{n \to \infty} \pi_{\text{real}}^{(n)}(B_n) = 0$.

Using the same analytical approach, we can obtain the ratio of maximum and minimum $p_{real}$ on $\bigcup_{i \leq m} \left\{d(\boldsymbol{x}, O_i) \leq \frac{1}{2}\sqrt{nw_j}\bar{d}_{\min}\right\}$: $\frac{C}{C_0}$ has some limit value. Combining with Equation (A93), we arrive at the conclusion that

$$\lim_{n \to \infty} \epsilon = \lim_{n \to \infty} \pi_{\text{real}}\left(\bigcup_{i \leq m}\left\{\boldsymbol{x} : d(\boldsymbol{x}, O_i) \leq \frac{1}{2}\sqrt{nw_j}\bar{d}_{\min}\right\}\right) \tag{A96}$$

$$= \lim_{n \to \infty} \frac{\pi_{\text{real}}\left(\bigcup_{i \leq m}\left\{\boldsymbol{x} : d(\boldsymbol{x}, O_i) \leq \frac{1}{2}\sqrt{nw_j}\bar{d}_{\min}\right\}\right)}{\pi_{\text{real}}(\text{supp}(\pi_{\text{real}}))} = 0. \tag{A97}$$

$\square$

# E EXPERIMENTAL SETTINGS

## E.1 EXPERIMENTS ON NUMERICAL CASES

**Settings for experiments in Section 3.1** To verify that instability indeed exists in diffusion generation, we conduct experiments on a two-dimensional diffusion model with a mixture of Gaussians as the generation distribution, consisting of three Gaussian components. To compute the intrinsic instability coefficient and obtain the results shown in Figure 3(b), we utilize the finite difference to estimate the intrinsic instability coefficient. The specific steps are as follows:

1. Uniformly sample initial points as on a $201 \times 201$ uniform grid of the area $[-1, 1] \times [-1, 1]$. Denote each initial point as $\boldsymbol{x}[i, j]$, where $i$ denotes the index along $x$-axis, and $j$ denotes the index along $y$-axis. Thus, $\boldsymbol{x}[i, j] = (-1 + \frac{i}{100}, -1 + \frac{j}{100})$ for $i, j = 0, 1, \ldots, 200$.

2. Numerically solve the PF-ODE in Equation (1) using the RK45 solver from $t = 1$ to $t = 0$. Each solution at $t = 1$, *i.e.*, the generated sample, can be denoted as $\hat{G}(\boldsymbol{x}[i, j])$ for each initial point $\boldsymbol{x}[i, j]$.

3. Estimate the intrinsic instability coefficient as $\mathcal{E}_G(\boldsymbol{x}[i,j], \boldsymbol{n}_y) \approx \|\hat{G}(\boldsymbol{x}[i, j + 1]) - \hat{G}(\boldsymbol{x}[i,j])\| / \|\boldsymbol{x}[i, j + 1] - \boldsymbol{x}[i,j]\|$.

For the instability coefficient $\mathcal{A}_F(\boldsymbol{x}, \boldsymbol{n}) := \frac{\|F(\boldsymbol{x}+\boldsymbol{n})-F(\boldsymbol{x})\|}{\|\boldsymbol{n}\|}$ in Figure 3(c), we use two points with larger input difference. As for its lower bound, we estimate it by leveraging the minimum intrinsic instability coefficient along the line segment connecting $\boldsymbol{x}$ and $\boldsymbol{x} + \boldsymbol{n}$. The specific steps are as follows:

1. Sampling points along the perturbation path. We uniformly sample 10 points along the line segment connecting $\boldsymbol{x}$ and $\boldsymbol{x} + \boldsymbol{n}$. Each point is represented as $\boldsymbol{x} + a \cdot \boldsymbol{n}$, where $a \in [0, 1]$. We chose 10 points to balance computational efficiency with accuracy for the estimation.

2. Estimating intrinsic instability at sampled points. At each of these 10 sampled points, we estimate the intrinsic instability coefficient $\mathcal{E}_G(\boldsymbol{x} + a \cdot \boldsymbol{n}, \boldsymbol{n})$ using a finite difference approximation with a step size of $\frac{1}{10}\|n\|$.

3. Calculating the lower bound. Let $\min\mathcal{E}_G$ denote the minimum value of $\mathcal{E}_G(\boldsymbol{x} + a \cdot \boldsymbol{n}, \boldsymbol{n})$ obtained from these 10 points. Then, the estimated lower bound for $\mathcal{A}_G$ is given by $\frac{2\sqrt{2}}{\pi}\min\mathcal{E}_G$.

It is important to note that this estimation method requires that the inner product of the Jacobian-vector products, $\langle J_F(\boldsymbol{x}_1)\boldsymbol{n}, J_F(\boldsymbol{x}_2)\boldsymbol{n}\rangle$, remains non-negative for any two points $\boldsymbol{x}_1, \boldsymbol{x}_2$ along the line segment connecting x and x+n. In our actual calculations for Figure Figure 3, we verified this condition using the finite difference estimations of the Jacobian-vector products at the 10 sampled points. Given the simplicity and smoothness of the Jacobian in the Figure 3 case, it was possible to choose a sufficiently small n to satisfy this condition.

The aforementioned calculation steps are supported by the following proposition:

**Proposition E.1.** *Suppose $F : \mathbb{R}^n \to \mathbb{R}^n$ is a continuously differentiable mapping. For $\boldsymbol{x} \in \mathbb{R}^n$, and a non-zero perturbation $\boldsymbol{n} \in \mathbb{R}^n$, if $\langle J_F(\boldsymbol{x} + a\boldsymbol{n})\boldsymbol{n}, J_F(\boldsymbol{x} + b\boldsymbol{n})\boldsymbol{n}\rangle \geq 0$ holds for any $a, b \in [0, 1]$, then we have*

$$\mathcal{A}_F(\boldsymbol{x}, \boldsymbol{n}) \geq \frac{2\sqrt{2}}{\pi}\min_a\mathcal{E}_F(\boldsymbol{x} + a\boldsymbol{n}, \boldsymbol{n}), \tag{A98}$$

*where $\mathcal{A}(\cdot, \cdot)$ is defined in Equation (3), $\mathcal{E}_F(\cdot, \cdot)$ is defined in Equation (2).*

This proposition essentially states that if the Jacobian-vector products along the perturbation path maintain a consistent directional relationship (non-negative inner product), then the overall amplification (instability coefficient) is lower-bounded by the minimum local amplification (intrinsic instability coefficient) along that path, scaled by a constant.

**Settings for experiments in Section 3.2** Similarly, we adopt the two-dimensional mixture of Gaussians as the generation distribution for these experiments. To obtain the correlation between reconstruction error and instability coefficient as shown in Figure 4(b), we follow the experimental procedure as below:

1. Uniformly sample initial data from $[-1, 1] \times [-1, 1]$.

2. Compute the reconstructed samples using the diffusion reconstruction process: first obtain the inverted noise $\hat{z} = \hat{G}^{-1}(\boldsymbol{x})$ for each initial data $\boldsymbol{x}$, and then regenerate the data as $\hat{\boldsymbol{x}} = \hat{G}(\hat{z})$.

3. Calculate the reconstruction error $\mathcal{R}(\boldsymbol{x})$ for each sample.

4. Estimate the intrinsic instability coefficient by 1) applying a small perturbation noise $\boldsymbol{n}$ to each inverted noise $\hat{z}$, and then 2) regenerating the data under perturbation as $\tilde{\boldsymbol{x}} = \hat{G}(\hat{z} + \boldsymbol{n})$, and 3) finally resulting in the estimation of intrinsic instability coefficient as $\mathcal{E}_G(\hat{z}, \frac{\boldsymbol{n}}{\|\boldsymbol{n}\|}) \approx \|\tilde{\boldsymbol{x}} - \hat{\boldsymbol{x}}\| / \|\boldsymbol{n}\|$.

5. Statistically analyze the correlation between $\mathcal{R}(\boldsymbol{x})$ and the intrinsic instability coefficient.

This procedure allows us to empirically assess the relationship between instability coefficients and reconstruction inaccuracies, thereby validating the theoretical insights discussed in Section 3.

## E.2 EXPERIMENTS ON TEXT-TO-IMAGE DIFFUSION MODELS

**Model and inference settings.** Models used in our experiments include Stable Diffusion 3.5 Medium / Large (Esser et al., 2024) and FLUX.1-dev (Black Forest Labs, 2024), accessed via the `diffusers` library (Dhariwal & Nichol, 2021) under the PyTorch framework. The inference is conducted on NVIDIA A800 GPUs.

During inference, most of the default scheduler configuration parameters are adopted for the numerical solution of the PF ODE. As for the ODE solver, we adopt Euler method following the default setting, except for experiments in Section B.1 that the Heun ODE solver is used. Besides, the number of inference steps is adjusted to 500 steps for both diffusion inversion and re-generation. The procedure to obtain the reconstruction error and the intrinsic instability coefficient is similar to that in the numerical experiments. The adopted image dataset is introduced below.

**Datasets.** The experiments primarily utilize the MSCOCO2014 dataset (Lin et al., 2014). For the experiments presented in Figure 4, we randomly select 100 images from the validation set of MSCOCO2014 for diffusion reconstruction. As for experiments in Section B.1, we adopt the same sampled images as in Figure 4.

## F ADDITIONAL RECONSTRUCTED IMAGES

To visually demonstrate that images are often difficult to be accurately reconstructed, we present failure cases of reconstruction by Stable Diffusion 3.5 (Esser et al., 2024). These images are selected from Kodak24 dataset (Franzen, R, 1999) in Figure A2 and MS-COCO 2014 (Lin et al., 2014) in Figure A3. For the reconstruction that includes both diffusion inversion and regeneration processes, we follow the default scheduler setting in diffusers (von Platen et al., 2022) with null text prompt and 100 inference steps for both inversion and regeneration.

## G LICENSES OF USED DATASETS

We list all the licenses of used datasets, code and models in Table A1.

Table A1: Licenses of datasets, code and models used in the paper.

| Name | License |
|---|---|
| **Datasets** | |
| MS-COCO2014 (Lin et al., 2014) | Creative Commons Attribution 4.0 License |
| Kodak24 (Franzen, R, 1999) | Free |
| **Code** | |
| Diffusers (von Platen et al., 2022) | Apache License 2.0 |
| **Models** | |
| Stable Diffusion 3.5 Medium (Esser et al., 2024) | Stability AI Community License |
| Stable Diffusion 3.5 Large (Esser et al., 2024) | Stability AI Community License |
| FLUX.1-Dev (Black Forest Labs, 2024) | FLUX.1 [dev] Non-Commercial License |

## H THE USE OF LARGE LANGUAGE MODELS

In this paper, LLM was used solely to refine text, syntax, and enhance readability. It did not contribute to anything related to the core ideas or scientific content (ideas, methods, theories, derivations, charts, results, and so on). In addition, all LLM-refined texts are manually double-checked for hallucinations and misunderstanding.

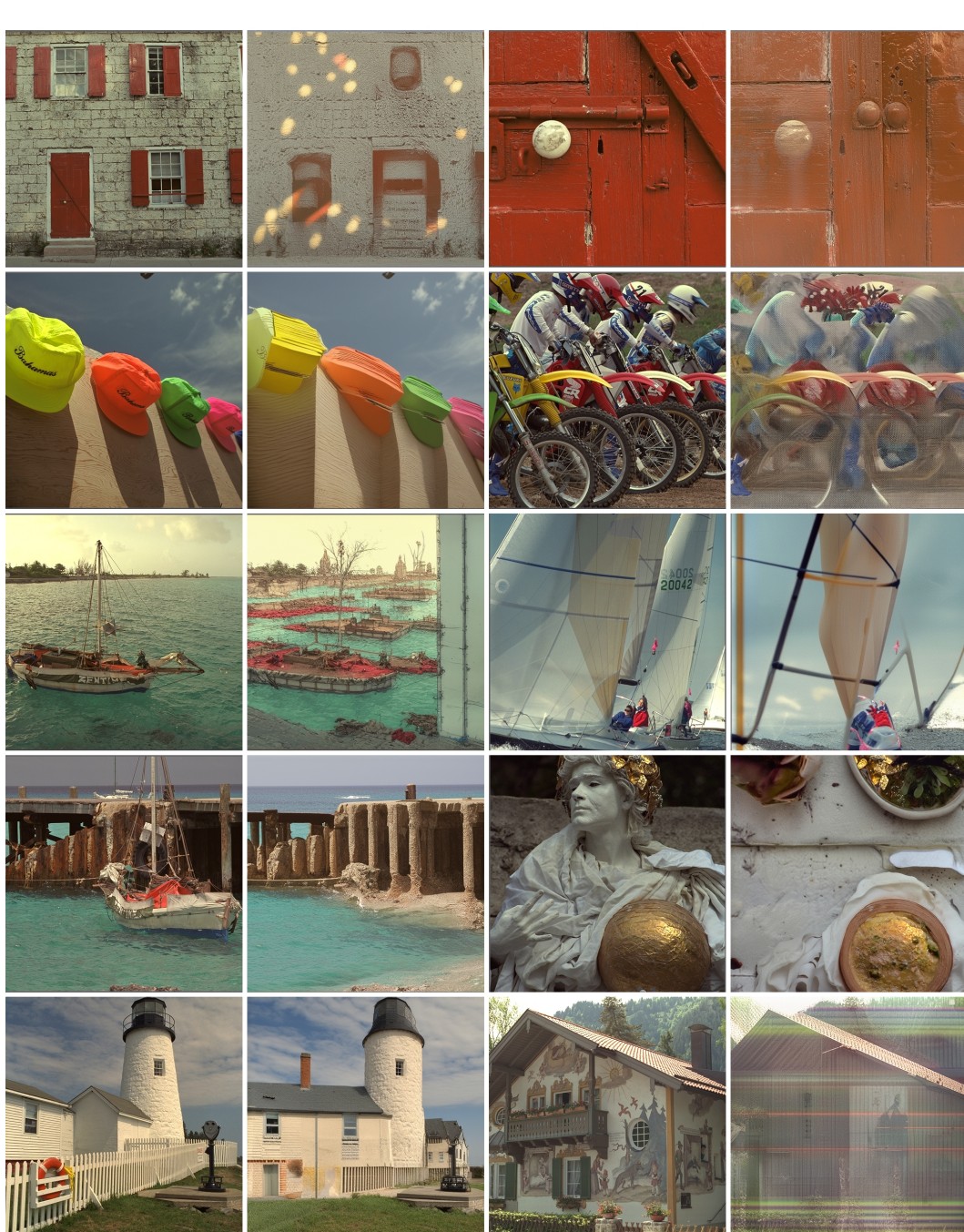

Figure A2: More failure cases of reconstruction on real images from Kodak24 dataset (Franzen, R, 1999) by Stable Diffusion 3.5 (Esser et al., 2024). In each row, the first and the third images are original real images, another two images are reconstructed ones.

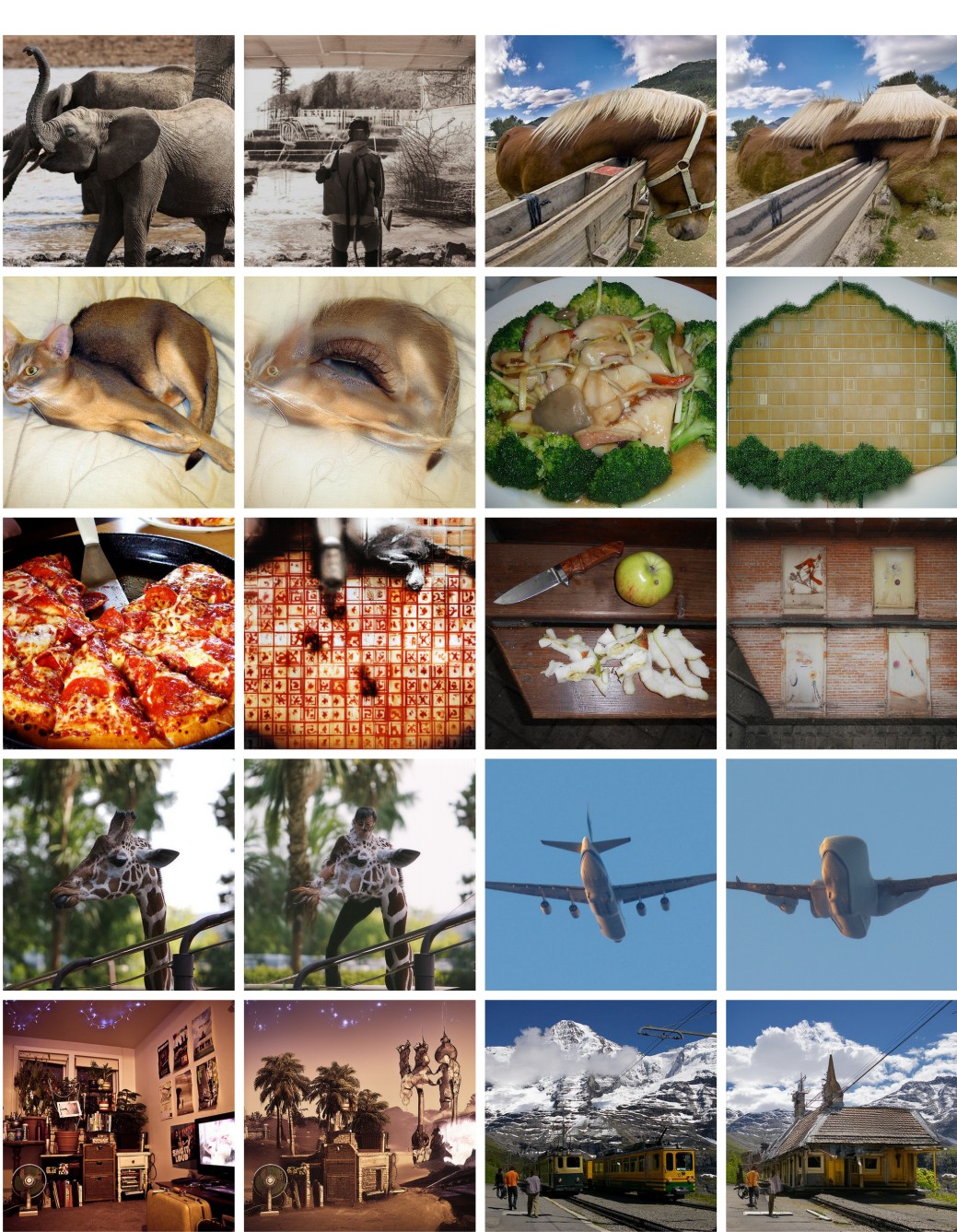

Figure A3: More failure cases of reconstruction on real images from MS-COCO 2014 datadset (Lin et al., 2014) by Stable Diffusion 3.5 (Esser et al., 2024). In each row, the first and the third images are original real images, another two images are reconstructed ones.

