# OpenReview forum: "Instability in Diffusion ODEs: An Explanation for Inaccurate Image Reconstruction"
_ICLR.cc/2026/Conference — Submitted to ICLR 2026_

### Official Review · Reviewer_PX99 · 2025-10-27

**Soundness:** 3
**Presentation:** 2
**Contribution:** 2
**Rating:** 2
**Confidence:** 3

**Summary:**

This paper focuses on analyzing the instability effect when performing round-trip image reconstruction in a diffusion or flow-matching model. A few new quantities such as "Intrinsic instability" and "intrinsic instability coefficient" are defined to facilitate the analysis. Instead of analyzing the instability at each time instant, the paper considers the instability of overall mapping function G(z) by solving the ODE over the entire time-interval.  A lower bound of instability’s probability is obtained. The paper further considers the instability of a diffusion model in an infinite dimensional space.

**Strengths:**

Overall, the papers provides an in-depth analysis for the instability effect of the round-trip image reconstruction in diffusion or flow-matching models.
(1) A lower bound of instability's probability is derived, which is based on the assumption of an ideal generation and inversion mapping. This indicates that the lower bound is independent of any particular ODE solver.

(2) The paper further considers the instability of a diffusion model for the extreme case of an infinite dimensional space under a number of assumptions. It shows that instability is inevitable in high probability.

**Weaknesses:**

(1) The demonstration in Figure 3 does not represent practical diffusion or flow-matching models. In practice, either variance-preserving or variance-exploding models are trained, where in the forward process, the variance of the noisy data samples either remains the same or increases as more noise is introduced to the clean data samples. In Figure 3 (a), it is clear that the variance of the clean data samples (from a GMM) is much larger than the variance of the pure Gaussian noise. As a result, the analysis in (b) and (c) for (a) does not indicate that the practice pre-trained  models have similar properties.

(2) Proposition 3.1 only considers the reconstruction error for the Euler method. Nowadays, advanced ODE solvers have been proposed that can perform both efficient sampling and exact reversibility. This makes the derived lower bound less informative.

(3) It feels that the assumptions for deriving Theorem 4.3 is a bit too strong. For example, Theorem 4.3 considers the extreme case that n approaches infinity while it requires m in Assumption 4.3 to be constant.  In practice, latent diffusion models have been proposed to reduce dimensionality in the latent space when performing iterative denoising. I highly suspect if the results in Theorem 4.3 could really provide any practice guidance in designing new generative AI models.

**Questions:**

Minor comments:

(1) There are several grammar errors in the paper, such as "In this section, provide theoretical evidence ...", and "To prove a almost sure".

---

### Official Review · Reviewer_G1T2 · 2025-10-30

**Soundness:** 3
**Presentation:** 2
**Contribution:** 3
**Rating:** 6
**Confidence:** 2

**Summary:**

This paper studies the intrinsic instability of diffusion ODEs and argues that reconstruction errors are not only from numerical discretization but also from the sparsity of the generation distribution, which amplifies small errors. The authors give theoretical proofs (including probabilistic lower bounds and asymptotic analysis) showing that such instability tends to occur almost surely in high-dimensional settings.

**Strengths:**

（1） The paper introduces a fresh and quite interesting perspective that instability caused by sparse generation distribution can explain inaccurate reconstruction.

（2）The theoretical analysis is clean and well-organized. Section 4 gives a solid chain of reasoning (Theorem 4.1–4.3) and makes the logic very clear.

**Weaknesses:**

（1）The experiments are a bit limited and mostly qualitative. The claim of “universal instability” would be stronger with comparisons across different diffusion models or solver variants.

（2）The paper has many formal definitions and assumptions but lacks discussion on practical implications. The theoretical idea is interesting, but it is not very clear how to use it in practice.

**Questions:**

（1）You argue that even small numerical and score approximation errors can be amplified by instability. Could you design an experiment with near-perfect scores and minimal numerical error to demonstrate that instability is the dominant factor?

（2）In Figure 4, correlation does not necessarily mean causation. Could it be that both instability and reconstruction error come from the same underlying issue, e.g., poor score approximation in low-density regions?

---

### Official Review · Reviewer_T1vq · 2025-10-30

**Soundness:** 3
**Presentation:** 1
**Contribution:** 1
**Rating:** 2
**Confidence:** 4

**Summary:**

This paper points out that 'instability' of the ODE itself is a reason for errors in diffusion model reconstruction  (image -> noise -> image). The authors argue that the 'sparsity' of the generation distribution causes this instability. Because of this, the numerical discretization error is amplified, and this leads to inaccurate image reconstruction The authors say they prove this with toy examples, real model experiments (SD 3.5), and a theory that the probability of instability becomes 1 as data dimension increases.

**Strengths:**

1. This paper tries to solve an important problem, which is the error in diffusion model reconstruction. This is important for many applications and many papers have studied this.

2. Figure 4 shows experimentally that the reconstruction error and the instability coefficient have a positive correlation. This supports the paper's claim.

**Weaknesses:**

1. Lack of Contribution (Obvious Claims): The main claim of the paper, "an unstable system (function with large Jacobian norm) amplifies input noise (numerical error)," is a very well-known and obvious. The paper names this 'instability', but it does not persuasively show why this is a new contribution in the context of diffusion models. Figure 4, which shows a correlation, also just re-confirms this obvious result.

2. The Propositions are very weak to support the logic. Proposition 2.1 is an almost obvious result from the definition of the Jacobian. It only explains the relation between finite perturbation and infinitesimal perturbation. It wastes space and gives no new insight. Moreover, in Proposition 3.1. This proposition claims that the 'upper bound (U) of the reconstruction error is greater than some value (C)'. A lower bound on an upper bound gives no useful information about the actual Error. (Error <= U and U >= C tells us nothing about Error.) For example, I can also make a claim "The upper bound of the error is greater than 157," which can be true but is an empty claim. This is fundamentally different from giving a lower bound for the Error itself, and it cannot be used as theoretical support.

3. The paper implicitly assumes the existence of an 'ideal' inverse mapping $G^{-1}$ (Definition 2.1). However, there is no discussion about whether the generation mapping $G$ is a bijection, so that $G^{-1}$ is uniquely defined, or if it is tractable. The $\hat{G}^{-1}$ (e.g., DDIM inversion or many other methods) used in actual reconstruction is just a numerical approximation of $G^{-1}$, so they are different. The paper calls the difference between $\hat{z}$ and $z_{ideal}$ (Figure 1) just 'numerical error'. But since the existence and uniqueness of $z_{ideal} = G^{-1}$ are not proven, this whole discussion is fundamentally vague.

4. Figure 2 is just an intuitive illustration for claiming Area(B) >> Area(A). The 2D toy example in Figure 3 is not appropriate for representing the complex manifold structure of high-dimensional image data.

5. I, in conclusion, think that papers about diffusion inversion accuracy, which only say the ODE is one-to-one but mathematically unstable without looking at how the actual numerical algorithm works, do not give new knowledge to researchers in this field. This is because researchers like Wallace, Pan, Hong, and Zhang already assume such things and have been working hard to figure out how to make a practical, numerical $G^{-1}$

### references
- Guoqiang Zhang, Jonathan P Lewis, and W Bastiaan Kleijn. Exact diffusion inversion via bi-directional integration approximation. arXiv:2307.10829, 2023.
- Zhihong Pan, Riccardo Gherardi, Xiufeng Xie, and Stephen Huang. Effective real image editing with accelerated iterative diffusion inversion. In ICCV, pages 15912–15921, 2023.
- Seongmin Hong, Kyeonghyun Lee, Suh Yoon Jeon, Hyewon Bae, Se Young Chun. On Exact Inversion of DPM-Solvers. In CVPR, 2024.

**Questions:**

Is 'Diffusion reconstruction' standard term in this field? I think 'inversion' is more prevail in prior work.

Why did you use geometric average to quantify the mapping's jacobian, among many types of matrix norm (of the singular value matrix)

---

### Official Review · Reviewer_z78t · 2025-11-05

**Soundness:** 2
**Presentation:** 1
**Contribution:** 2
**Rating:** 2
**Confidence:** 3

**Summary:**

The paper investigates instability in diffusion ODEs. Specifically, the authors highlight the issue of inaccurate reconstructions where, despite the ODEs defining perfectly invertible mappings, in practice applying the forward and reverse process may result in notable inaccuracies. Fundamentally, the authors attribute this to the "sparsity inherent in the generation distribution: the probability mass is concentrated on scattered small regions, while most of the space remains nearly empty". The paper provides some experimental evidence and also some theoretical derivations, implying that inaccuracies are amplified in higher-dimensional data.

**Strengths:**

1. The paper attempts to rigorously understand the instability phenomenon via a mathematical treatment.
2. The visualizations (mostly given in the Appendix) are convincing of the issue.
3. To my knowledge, the analysis on instability as a function of dimensionality is novel. However, I have not carefully checked the proofs for correctness.

**Weaknesses:**

1. Related work should be more carefully discussed. The instability phenomenon has been previously studied for general invertible networks (e.g., see [1]) and it is well understood that non volume perserving mappings are inherently more prone to this. In this sense, the novel contribution of the paper is unclear.

2. The writing is imprecise. For example:

    - The term "sparsity" is used to mean, as I understand, support on low-dimensional manifolds. This leads to confusing statements like: "training samples are sparse" (line 400); "Considering the sparsity of high-dimensional images" (line 404).

    - "sparsity in the generation distribution" does not reflect the underlying issue, which is the spectrum of the Jacobian / volume transformations and associated Lipschitz constants of the overall generation process.

3. The experiments are quite minimal and not convincing. For example, I see no empirical validation for the analysis of Section 4.

4. Proposition 2.1 appears weak in the sense that the bound is loose and n is unconstrained. It is therefore unclear what A_F actually measures.

**Questions:**

1. There are schemes in the literature (e.g., [2]) that achieve perfect inversion and up to machine precision (by compensation for accumulation errors). I expect that these methods work regardless of the proposed analysis so perhaps the inversion issues discussed in this paper can be circumvented. Can the authors comment on this?

2. Could the authors include an experiment that supports Section 4? For example, the authors can include reconstructions of the same image at different resolutions.

3. Line 774 and below claims that instability shows generalization limitations of diffusion models and because of it they may not learn the true data distribution. Why is numerical discretization not sufficient to explain reconstruction error (line 47)? Intuitively, instabilities emerge during ODE inference due to numerical errors, not during the training / learning process as this focuses solely on the score function in isolation from sampling dynamics. Note, I saw your solver agnostic analysis in Section 4. However, as I understand, P_M tending to 1 (Theorem 4.3) is a statement about an underlying Jacobian matrix whose spectrum is governed by the associated score function, not the actual errors. The fact that your theorem appears applicable even if one substitutes in the true score function suggests that instability persists even when models optimally learn the scores and therefore the data distribution.

4. Proposition 3.1 gives a lower bound on the upper bound, "U", of reconstruction error. How do you define this "U"? Why is lower bounding an upper bound significant / meaningful?

[1] On the Invertibility of Invertible Neural Networks, 2020

[2] An Edit Friendly DDPM Noise Space: Inversion and Manipulations, 2024

---

### Official Review · Reviewer_ZxCZ · 2025-11-10

**Soundness:** 2
**Presentation:** 2
**Contribution:** 1
**Rating:** 2
**Confidence:** 4

**Summary:**

Diffusion-based reconstruction often leads to non-negligible reconstruction errors in practice, which can hinder the application of diffusion models on downstream tasks. This paper argues that these errors stem from instability in the generative mapping. Such instability originates from the sparsity of the generative distribution and amplifies the deviations introduced by the inversion process. This paper presents both theoretical and experimental analyses to demonstrate that instability amplifies reconstruction error, and that the probability of instability converges to one as data dimensionality grows.

**Strengths:**

- The paper provides a definition of instability and analyzes its influence on reconstruction errors using both experimental and theoretical analyses.

**Weaknesses:**

The primary weakness of this paper lies in the lack of a clear, reasonable connection between the so-called “instability” and “reconstruction error” observed in diffusion models. Moreover, much of the discussion appears casual, with insufficient references, weak theoretical grounding, and inaccurate mathematical descriptions.

- This paper attempts to explain reconstruction errors beyond numerical discretization error, attributing them to the concept of “instability” caused by the sparsity of the probability distribution. However, the connection between the defined “intrinsic instability” (Definition 2.2) and distribution sparsity is unclear. In fact, this definition that describes the local sensibility of a dynamic system is indeed more relevant to explaining how an advanced numerical solver can reduce the numerical discretization errors. In addition, these mathematical definitions and proofs are largely standard results from nonlinear analysis and dynamic systems, yet this paper does not include proper references.

- Lines 72-76 state: “Note that we analyze the ideal diffusion reconstruction process, free from numerical errors, to isolate properties of PF-ODE reconstruction that do not depend on the choice of numerical solvers”. However, this setting seems questionable. If numerical errors are completely eliminated, PF-ODE reconstruction becomes exact, and thus the reconstruction error problem effectively disappears in this idealized setting.

- Figure 2 illustrates that in a bijective probability transformation, if B falls into a low-density region than A, then Area(B)>Area(A). This is a trivial result and does not require a dedicated figure and lengthy caption to explain. Furthermore, this is unrelated to the case of diffusion models in practice, since a well-trained model rarely maps Gaussian noise to low-density regions. One simple example is that empirically, a sufficiently trained diffusion model produces a visually plausible output from almost any initial noise.

- This paper lacks deeper insights into the phenomenon of inaccurate reconstruction and does not propose or validate any improvements based on the presented analysis.

**Questions:**

- The term "large gradients in the generation process" (Lines 89-90) is mathematically imprecise and confusing. In diffusion models, the gradients in generation are usually referred to as the score function. It is unclear why this quantity would “change” in the sense implied by the authors.

- Lines 88-89 claim that instability roots in the generation process mapping a region A to a significantly lower-density region B. The authors are required to provide some experimental results to verify that the images with poor reconstruction quality indeed correspond to regions with lower probability density under the generation distribution.

---

### Meta-Review · Area_Chair_mUWM · 2025-12-31

**Summary:**

Reviewers highlighted a number of key concerns. These include a lack of novelty of weak contribution, unclear / poorly justified theoretical framework, insufficient / unconvincing experimental results, and clarity issues with the writing.

**Reviewer Concerns:**

There were no rebuttal comments nor any apparent modifications to the paper. Hence all reviewer concerns remain outstanding.

**Reviewer Scores:**

The original reviewer scores were 6, 2, 2, 2, 2 from five reviews. Given the lack of author responses, it is likely that all reviewers will unanimously recommend to reject the submission.

---

### Decision · Program_Chairs · 2026-01-26

Reject